# Comparison of Satellite Precipitation Products: IMERG and GSMaP with Rain Gauge Observations in Northern China

**Huiqin Zhu [1,2,3]**, **Sheng Chen [1,2,4,\*]**, **Zhi Li [3]**, **Liang Gao [5]** and **Xiaoyu Li [6]**

1   Key Laboratory of Remote Sensing of Gansu Province, Northwest Institute of Eco-Environment and Resources, Chinese Academy of Sciences, Lanzhou 730000, China
2   Heihe Remote Sensing Experimental Research Station, Northwest Institute of Eco-Environment and Resources, Chinese Academy of Sciences, Lanzhou 730000, China
3   School of Geography and Planning, Nanning Normal University, Nanning 530001, China
4   Southern Laboratory of Ocean Science and Engineering, Zhuhai 519000, China
5   State Key Laboratory of Internet of Things for Smart City, Department of Civil and Environmental Engineering, University of Macau, Macau 999078, China
6   School of Geography and Tourism, Jiaying University, Meizhou 514015, China
\*   Correspondence: chensheng@nieer.ac.cn; Tel.: +86-13802424815

**Abstract:** Extreme precipitation events have increasingly happened at global and regional scales as the global climate has changed in recent decades. Accurate quantitative precipitation estimation (QPE) plays an important role in the warning of extreme precipitation events. With hourly rain gauge observations as a reference, this study compares the performance of Integrated Multi-satellitE Retrievals for Global Precipitation Measurement (GPM) mission (IMERG) and Global Satellite Mapping of Precipitation (GSMaP) quantitative precipitation estimation (QPE) products over Northern China in 2021. The Probability of Detection (POD), Relative Bias (RB), Root-Mean-Squared Error (RMSE), and Fractional Standard Error (FSE) are among the assessment metrics, as are the Probability of Detection (POD), False Alarm Ratio (FAR), and Critical Success Index (CSI). We examined the spatial distribution of cumulative precipitation and the temporal distribution of hourly average precipitation for three severe precipitation occurrences using these assessment metrics. The IMERG products capture strong precipitation centers that are compatible with the gauge observations, especially in extreme precipitation events in areas with relatively flat terrain and low-altitude ($\leq$1000 m). Both IMERG (National Aeronautics and Space Administration, NASA) and GSMaP (Japan Aerospace Exploration Agency, JAXA) satellite-based QPE products have precipitation peaks in advance (2–4 h) and generally underestimate (overestimate) precipitation when the actual precipitation is heavy (light). The satellite-based QPE products generally overestimate the heavy rainfall caused by non-typhoons and underestimate the heavy rainfall caused by typhoons. The GSMaP products may have the capacity to detect short-term rainstorm events. The accuracy of satellite-based QPE products may be influenced by precipitation intensity, sensors, terrain, and other variables. Therefore, in accordance with our recommendations, more ground rainfall stations should be used to collect actual precipitation data in regions with high levels of spatial heterogeneity and complex topography. The data programmers should strengthen the weights computation retrieval technique and fully utilize infrared (IR)-based data. Furthermore, this study is expected to give helpful feedback to the algorithm developers of IMERG and GSMaP products, as well as those researchers into the use of IMERG and GSMaP satellite-based QPE products in applications.

**Keywords:** Northern China; GPM; GSMaP; IMERG; uncertainty; evaluation; extreme precipitation

## 1. Introduction

Under a changing climate, both the frequency and intensity of precipitation events tend to increase in many regions [1,2]. South Korea and Japan were hit by continuous heavy precipitation in early July 2021, which led to landslides and mass evacuations [3,4].

In mid-July 2021, many countries in western Europe were hit by heavy rains, which caused severe flooding, with the worst affected being Germany, where at least 177 people lost their lives in the floods [5]. Meanwhile, many parts of India and Nepal were affected by heavy precipitation, resulting in many floods and landslides [6].

Northern China has also experienced a series of extreme precipitation events. From 10–13 July 2021, Northern China, especially the Beijing–Tianjin–Hebei area, witnessed the first extreme precipitation event in the flood season. A total of seven national rainfall stations in central and southern Hebei and northern Henan exceeded the July extreme values, with Jize (206.4 mm) in Hebei and Huaxian (211.7 mm) in Henan breaking through the historical records [7]. Northern China, especially in Henan Province, which was mainly affected by typhoon In-Fa and typhoon Cempaka [8,9] had an extreme precipitation event on 20 July 2021. The majority of Henan Province suffered from a heavy rainfall storm from the 18–21 July 2021, with torrential rain occurring in Zhengzhou, Xuchang, and Xinxiang [10]. The hourly precipitation and daily precipitation have surpassed the historical records of the last 60 years since the foundation of Zhengzhou Station in 1951 [11]. In addition, Northern China experienced an extreme precipitation event that started at UCT (Coordinated Universal Time) 00:00 on 2 October 2021 and lasted five days until it ended at UTC 23: 00 on 6 October 2021 with maximum daily precipitation on 5 October. The 5 October Northern China Extreme Rainstorm is another extreme precipitation event (hereafter, 5 October NCER) that occurred in China after the 20 July extreme precipitation event in Northern China. The 5 October NCER was accompanied by lightning and strong convective weather in the early stage of the weather process, and the temperature decreased sharply and continued to decline in the later stage of the weather process [12]. The daily precipitation of 59 national meteorological stations in Shanxi Province broke through the historical extreme value of the same period since the establishment of the station, and the accumulated precipitation of 63 national meteorological stations exceeded the historical extreme value of the same period [12].

Wang et al. [13] investigated the distribution and trend of extraordinary precipitation in China, finding that extreme precipitation events have become more concentrated and intense in recent decades. Since extreme precipitation events have serious social and economic consequences, forecasting extreme precipitation events is particularly important. Currently, there are two methods for detecting precipitation: in-situ observation and remote sensing retrieval. In-situ observation instruments include ground rain gauges, and remote sensing retrieval includes ground weather radar and satellite precipitation estimation. In general, the rainfall data obtained by in-situ observation can provide the most accurate precipitation data, but this method does not perform well in terms of spatial continuity. The ground weather radar has a high spatial and temporal resolution, but it is susceptible to the impact of mountains on detection, resulting in data divergence. Satellite-based QPE products provide greater coverage than radar quantitative precipitation measurements, making them particularly useful for precipitation monitoring in mountainous locations, deserts, and seas. Due to the advantages of a large detection range, strong spatial continuity, and high spatial and temporal resolution, the satellite can quickly capture mesoscale heavy rainfall.

At present, the satellite-based QPE products play an important role in meteorological forecasting and disaster warning monitoring. Most researchers have focused on assessing the accuracy of satellite-based QPE products in extreme precipitation events [14–19]. Meanwhile, with the gradual improvement of satellite precipitation product evaluation, the factors that affect the accuracy of satellite precipitation product evaluation are gradually explored [20–22]. To date, many researchers agree that the main error sources of satellite-based QPE products in predicting precipitation are the configuration of satellite-based QPE products (sensors and algorithms for data processing) [23–25] and the geographical conditions of the regions (topography and altitude) [26–29]. Reviewing previous studies, some scholars also found that the accuracy of satellite precipitation products was also closely related to precipitation intensity [20,30–32]. According to the studies on the performance

of satellite-based QPE products by Saber [15], Zhang [33], Kim [34], and Chen [35], it is found that the distribution density of surface rain gauges is one of the important factors affecting the accuracy of products. This is because there is a lack of sufficient rain gauge observations to evaluate the objective performance of satellite-based QPE products over regions where the rain gauges are sparsely distributed.

Many efforts have been made to evaluate the GPM products (IMERG or GSMaP). The GPM mission is built upon the Tropical Rainfall Measuring Mission (TRMM) and can provide a global coverage of rain and snow products within 3 h based on microwave and 0.5 h based on infrared observations, and the detection range extends to the Arctic Circle and Antarctic Circle. The GPM satellite group is composed of the GPM core observatory launched by the National Aeronautics and Space Administration (NASA) and Japan Aerospace Exploration Agency (JAXA), the French National Centre for Space Studies (CNES), the European Meteorological Satellite (EUMETSAT), the Indian Space Research Organisation (ISRO), the American National Oceanic and Atmospheric Administration (NOAA), and United States Department of Defense (DOD) satellites. As far as we know, recent studies overseas generally focus on the long-term series assessment of satellite-based QPE products [36–40], and few studies have examined the performance of satellite-based QPE products in short-term emergencies. At the moment, many scholars are primarily interested in evaluating the temporal and spatial distribution characteristics of long-term series of meteorological events, such as drought and precipitation, using satellite-based QPE products at daily-scale data or monthly-scale data as basic research data [41–44]. Moreover, most studies employ different interpolation algorithms to interpolate rainfall station data and then use the interpolated data as the primary data for comparative analysis [45,46]. Various factors influence the interpolation process and hence have an impact on the actual outcomes. Furthermore, daily- and monthly-scale data cannot consistently and promptly anticipate real-time extreme precipitation occurrences, since extreme precipitation storms are quick and powerful. However, most of the assessments of satellite-based QPE products in China are concentrated in the large-scale regional areas of China [47–49] or the economically developed areas along the eastern coast, which are severely affected by typhoons [17–19,50]. Also, a lot of research is being performed right now to evaluate satellite-based QPE products depending on the river basin region by utilizing the hydrologic data from hydrological stations [51–54]. There are few studies on the accuracy of V06 IMERG and V07 GSMaP in the range of inland regions in Northern China, especially for comparing the extreme precipitation events not originating from typhoons with those originating from typhoons.

In this study, with the data of two extreme precipitation events generated by non-typhoons and one extreme precipitation event caused by a typhoon, we review and analyze three extreme precipitation events that happened in Northern China in 2021. For these extreme precipitation events in Northern China, the study takes the ground rain gauge observations as a reference, and adopts the Correlation Coefficient (CC), Relative Bias (RB), Root-Mean-Squared Error (RMSE), and Fractional Standard Error (FSE), as well as the Probability of Detection (POD), False Alarm Ration (FAR) and Critical Success Index (CSI) to evaluate data from the Version 06 of the GPM mission, developed by NASA IMERG, and a new satellite precipitation product developed by JAXA the GSMaP in Version 07.

This study provides a detailed analysis of the uncertainty characteristics of satellite-based QPE products in the context of the Northern China's extreme rainstorms in 2021, and evaluates the performance of IMERG and GSMaP satellite-based QPE products. This paper attempts to help address the uncertainties and inaccuracies in satellite-based QPE products' future development, provides helpful feedback to algorithm developers, and affords a reference for improving the accuracy of satellite-based QPE products, popularization and application, and subsequent prediction of extreme precipitation events on a regional scale.

The rest of this paper is organized as follows. Section 2 introduces the study area, the rain gauge observations data, satellite-based QPE products dataset, and evaluation metrics. Section 3 focuses on the analysis of time and spatial characteristics and contingency

information for satellite-based QPE products. Section 4 focuses on analyzing the causes of satellite-based QPE products' uncertainty. The conclusions are briefly summarized in Section 5.

## 2. Materials and Methods

### 2.1. Study Area

The study area is located in Northern China, which was affected by the 12 July 2021 Northern China Extreme Rainstorm, by the 20 July 2021 Northern China Extreme Rainstorm, and by the 5 October 2021 Northern China Extreme Rainstorm in 2021 (hereafter referred to as 12 July NCER, 20 July NCER and 5 October NCER). These spanned from 31°N to 43°N in latitude and from 105°E to 123°E in longitude (Figure 1). The topography of the study area varies significantly in different regions (Figure 1a). Under the combined action of climatic conditions and topographic factors, the Northern China disasters have the features of high frequency and high intensity. With frequent gales, droughts, rainstorms, and blizzards occurring in this region, Northern China is facing weather disasters with great frequency of weather disasters. Simultaneously, floods and landslides caused by heavy precipitation are especially severe.

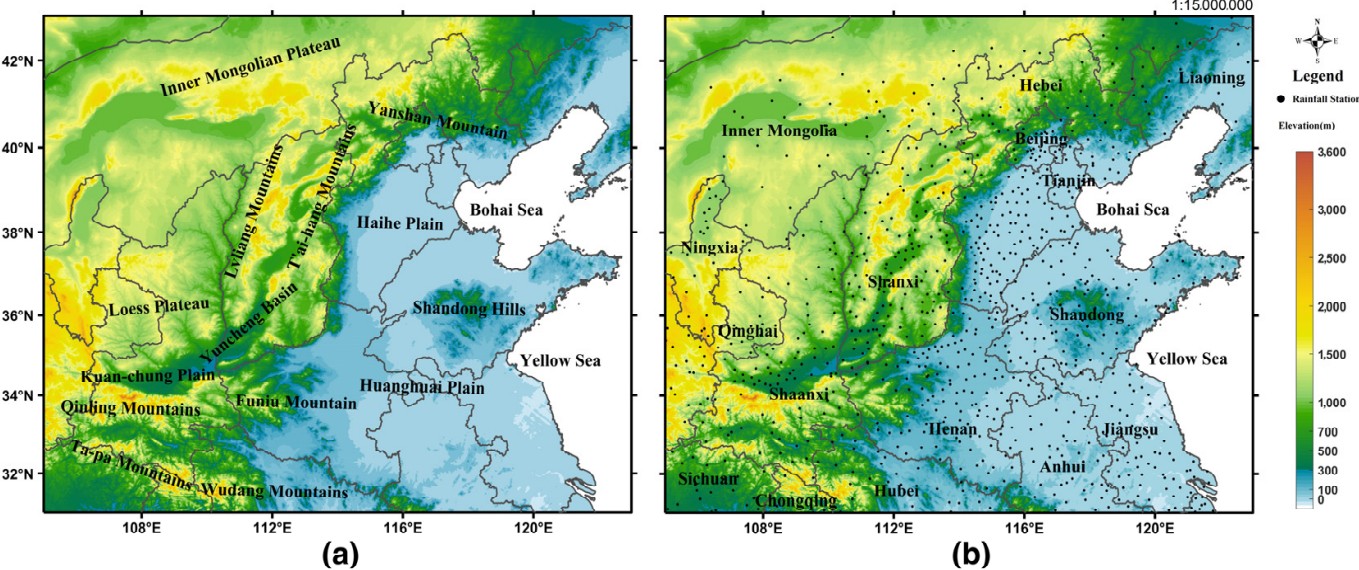

**Figure 1.** The topography of study area (**a**) and distribution of ground rainfall gauge stations (**b**).

### 2.2. Datasets

#### 2.2.1. Rain Gauge Observations

The reference data are hourly observations from surface observation stations in China and were obtained from the China Meteorological Data Service Center (CMDC, http://data.cma.cn/, accessed on 7 October 2021), with the 12 July NCER spanning from UTC 00:00 on 10 July 2021 to 23:00 on 13 July 2021, the 20 July NCER from UTC 00:00 on 17 July 2021 to 23:00 on 22 July 2021, and the 5 October NCER from UTC 00:00 on 2 October 2021 to 23:00 on 6 October 2021. The distribution of 906 national-level surface observation stations in the study area is shown in Figure 1b. In this paper, the number of surface observation stations employed in three extreme precipitation occurrences varies: 557 on 12 July NCER, 535 on 20 July NCER, and 831 on 5 October NCER. Precipitation observation data can accurately reflect the basic precipitation situation near the rainfall station [55]. All the reference data are quality controlled by the National Meteorological Information Centre (NMIC), such as the station or regional extremes check and the temporal and spatial consistency checks [56–58].

2.2.2. Satellite Data

IMERG is a new generation of multi-satellite merged precipitation retrieval products. It makes full use of the data provided by active and passive microwave sensors and various infrared sensors on the GPM platform, as well as the satellite precipitation retrieval algorithm in the era of the TRMM. IMERG employs passive microwave (PMW) sensors to obtain precipitation estimates. Currently, IMERG provides three types of satellite precipitation products: Early Run, Late Run, and Final Run, which all have three temporal resolutions (Half hour, 1 day, and 1 month). In version 6.0, three satellite precipitation products all have three sub-products with the same spatial and temporal resolutions: $0.1°/30$ min, $0.1°/1$ day, and $0.1°/1$ month (the spatial resolution between 60°N and 60°S in previous versions) [25,59]. The Final Run product has been corrected by the global rain gauge analysis product, which is better than the Early Run and Late Run products in terms of accuracy, but is usually released about 3.5 months later than the Early Run and Late Run [25,59]. Early Run and Late Run are quasi-real-time products. The delay time of Early Run and Late Run is with a short delay of 4 h and 14 h, respectively. The Early Run uses only forward morphing algorithms, while the Late Run applies both forward and backward morphing algorithms [25,59].

GSMaP is a global precipitation product released by JAXA that provides high-precision and high-resolution data recorded by passive microwave radiometers (MWRs) and geostationary infrared radiometers (Geo-IRs) [60]. GSMaP products are calculated using data from several microwave radiometers and a rain rate retrieval algorithm based on a credible rainfall physics model. GSMaP products were generated for the GPM mission based on the GSMaP project's activities. The Global Rainfall Map in the Near-Real-Time product (GSMaP_NRT) is derived from the precipitation rate from GSMaP_MWR (Global Satellite Mapping of Microwave Radiometer) [60] by employing the microwave sounder algorithm, microwave imager algorithm, and microwave–IR combined algorithm to modify the precipitation datasets [20]. The GSMaP standard version GSMaP_MVK (Global Satellite Mapping of Precipitation Microwave –IR Combined Product) makes use of microwave–infrared algorithms and adopts both forward and backward propagation processes [61]. The GSMaP_Gauge (Gauge-calibrated Rainfall Product) is a product that regulates the GSMaP_MVK estimate with the global rainfall gauge analysis, climate and topography by United States CPC (Climate Prediction Center) [62]. The spatial and temporal resolutions of GSMaP precipitation products are $0.1°/1$ h (within 60°N–60°S), and GSMaP_NRT delays 4 h, while GSMaP_MVK and GSMaP_Gauge delay 3 days.

Considering the need for real-time data accuracy assessments and to facilitate comparison between the two kinds of half-hour GPM precipitation products, the study uses the quasi-real-time (Early Run) and near-real-time (Late Run) data from IMERG Version 06. This was provided by the NASA website (https://gpm.nasa.gov/, accessed on 25 October 2021), namely IMERG_ERUnCal and IMERG_LRUnCal (hereafter referred to as IMERG_ER and IMERG_LR), and GSMaP_NRT and GSMaP_MVK data from JAXA's website (http://www.gportal.jaxa.jp/, accessed on 25 October 2021) in GSMaP Version 07.

The primary differences and similarities characteristic of satellite-based QPE products that are utilized in this article are summarized in Table 1.

**Table 1.** The major differences and common features of satellite-based QPE products.

| Product | Spatial/ Temporal Resolution | Spatial Domain | Main Input Data | Latency | Applications |
|---|---|---|---|---|---|
| IMERG_ER | 0.1°/0.5 h | 90°N–90°S | PMW, DPR [1], GMI [2], PR [3], TMI [4], AMSR2 [5], SSMIS [6] | 4 h | Prediction of flash floods and precipitation |
| IMERG_LR | 0.1°/0.5 h | 90°N–90°S | PMW, DPR, GMI, PR, TMI, AMSR2, SSMIS | 14 h | Water resource management |
| GSMaP_NRT | 0.1°/1 h | 60°N–60°S | TMI, GMI, SSMIS, PR, DPR, AMSU-A [7]/AMSU-B [8] | 4 h | Real-time applications |
| GSMaP_MVK | 0.1°/1 h | 60°N–60°S | TMI, GMI, SSMIS, PR, DPR, AMSU-A/ AMSU-B | 3 days | Water resource management |

[1] GPM Dual-frequency Precipitation Radar; [2] GPM Microwave Imager; [3] TRMM Precipitation Radar; [4] TRMM Microwave Imager; [5] Advanced Microwave Scanning Radiometer Version 2; [6] Special Sensor Microwave Imager/Sounder; [7] Advanced Microwave Sounding Unit A; [8] Advanced Microwave Sounding Unit B.

### 2.2.3. Initial Processing

To evaluate the performance of IMERG and GSMaP satellite-based QPE products over Northern China in 2021, the hourly gauge observations (called Gauge, referred to as GG) were used as a reference. Since the data types and spatial–temporal scales of GG and satellite precipitation data are different, it is necessary to preprocess these two kinds of data for fair comparison. For the satellite data, the half-hourly IMERG product is accumulated into hourly data. The gauge observations are compared with the IMERG and GSMaP products for grids that were overlapped with at least one gauge. Kriging interpolation is a technique that is widely used in geoscientific applications such as precipitation [63,64], soils, and ground wind fields. Using Kriging can produce unbiased predictions with the smallest variance and examine the spatial correlation between the data gathered by various rain gauges or meteorological stations [65–67]. The GG were interpolated into regular gridded analysis using Kriging Interpolation in Interactive Data Language (IDL, Version 8.5), with a spatial resolution of 0.1° × 0.1° in this research (referred to as GGKRIG). The kriging interpolation may increase the uncertainty of gridded gauge analysis, but this study neglects these uncertainties for spatial analysis of GG and satellite-based QPE products. The GGKRIG was only used for the display of satellite-based QPE products and was not utilized for further statistical verification. During the data processing process, it was discovered that 12 h of GSMaP_MVK data (UTC 00:00 to 01:00 on 4 October 2021, UTC 03:00 on 4 October 2021, UTC 11:00 on 5 October 2021, UTC 13:00 to 14:00 on 5 October 2021, and UTC 10:00 to 15:00 on 6 October 2021) have problems for 5 October NCER (Figure 2a). To enhance the accuracy and fairness of the examination, all satellite-based QPE products data are omitted from this period of data.

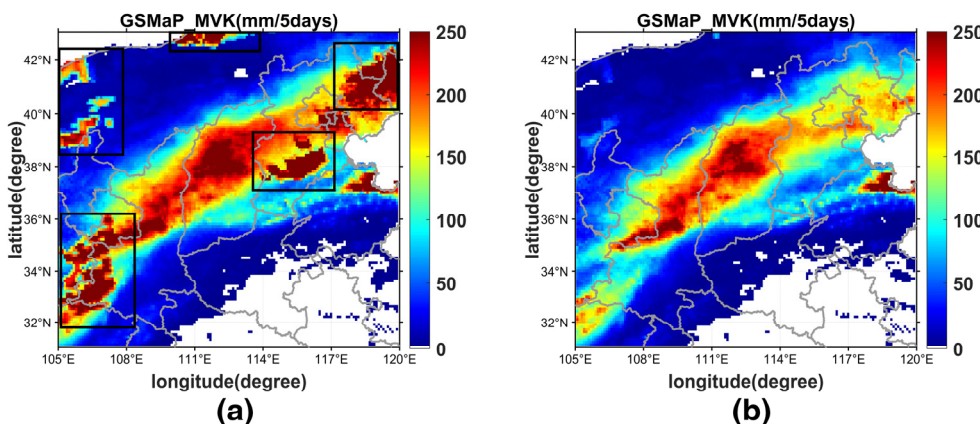

**Figure 2.** Distribution of precipitation from 5 October NCER GSMaP_MVK (**a**) before and (**b**) after quality control. The black rectangle indicates the area with abnormal rainfall estimation.

### 2.3. Statistics Metrics

In this study, gauge observations are used as a reference to evaluate the performance of version 06 IMERG products and version 07 GSMaP products. Seven skill indices and statistical methods are used to assess the accuracy of these two kinds of satellite products. Table 2 lists the statistic indices for the CC, RB, RMSE, and FSE scores, as well as the POD, FAR, and CSI. The spatial and temporal correlation between satellite precipitation products and rain gauges is represented by CC. RB describes the error trend between satellite precipitation products and gauge observations. The RMSE indicates the overall accuracy and error level of IMERG and GSMaP precipitation estimates. The FSE shows the difference in average precipitation data between satellite precipitation products and ground rainfall stations, and it is used to quantify the accuracy of IMERG and GSMaP precipitation predictions. The capacity of satellite precipitation products to catch real precipitation occurrences is referred to as POD. FAR is the degree of false alarm of IMERG and GSMaP precipitation estimates for precipitation events. CSI examines the full examination of POD and FAR, which identifies the genuine amount of satellite-based QPE products to identify actual precipitation occurrences.

**Table 2.** Satellite precipitation product evaluation indicators.

| Statistic Indices | Formula | Range | Optimum Value |
|---|---|---|---|
| Correlation Coefficient (CC) | $\dfrac{\sum_{i=1}^{n}(G_i-\overline{G})(S_i-\overline{S})}{\sqrt{\sum_{i=1}^{n}(G_i-\overline{G})^2}\times\sqrt{\sum_{i=1}^{n}(S_i-\overline{S})^2}}$ | $[-1, 1]$ | 1 |
| Relative Bias (RB) | $\dfrac{\sum_{i=1}^{n}(S_i-G_i)}{G_i}\times 100\%$ | $(-\infty, \infty)$ | 0 |
| Root-Mean-Squared Error (RMSE) | $\sqrt{\dfrac{1}{n}\sum_{i=1}^{n}(S_i-G_i)^2}$ | $[0, \infty)$ | 0 |
| Fractional Standard Error (FSE) | $\sqrt{\dfrac{\frac{1}{n}\sum_{i=1}^{n}(S_i-G_i)^2}{Avg(G_i)}}$ | $[0, \infty)$ | 0 |
| Probability of Detection (POD) | $\dfrac{A}{A+C}$ | $[0, 1]$ | 1 |
| False Alarm Ration (FAR) | $\dfrac{B}{A+B}$ | $[0, 1]$ | 0 |
| Critical Success Index (CSI) | $\dfrac{A}{A+B+C}$ | $[0, 1]$ | 1 |

$n$ denotes the total number of grid points; $S_i$ represents the precipitation estimated by satellite-based QPE products; $G_i$ indicates the value of gauge observations interpolated to the grid point; $\overline{S}$ signifies the average of satellite-based QPE products' estimated precipitation; $\overline{G}$ represents the average precipitation value after interpolation of the gauge observations; $Avg(G_i)$ denotes the average value of the gauge observations interpolated to the grid; $A$ shows the number of precipitation events detected by the satellite-based QPE products; $B$ indicates the number of precipitation events missed by satellite-based QPE products; $C$ represents the number of precipitation events misreported by the satellite-based QPE products.

The decomposition model of overall error components of satellite precipitation presented by Tian et al. [30] is frequently used to assess the detection capabilities of satellite-based QPE products to precipitation events. Among them, POD, CSI, and FAR are used to compare the performance of IMERG and GSMaP satellite precipitation products under different precipitation thresholds. The Chinese precipitation magnitude levels and their corresponding thresholds are shown in Table 3 [68], and the World Meteorological Organization (WMO) standard of daily precipitation is listed in Table 4 [69].

**Table 3.** Chinese precipitation magnitude and its corresponding thresholds.

| Precipitation Grade | Daily Precipitation (mm) |
| --- | --- |
| Light Rain | $1 \leq i < 10$ |
| Moderate Rain | $10 \leq i < 25$ |
| Heavy Rain | $25 \leq i < 50$ |
| Rainstorm | $50 \leq i < 100$ |
| Large Rainstorm | $100 \leq i < 250$ |
| Extraordinary Rainstorm | $i \geq 250$ |

Avoiding the effect of micro-precipitation, adjust the minimum level of light rain to 1 mm/d; i represents the daily precipitation.

**Table 4.** World Meteorological Organization (WMO) standard of daily precipitation.

| Precipitation Grade | Daily Precipitation (mm) |
| --- | --- |
| Light Rain | $i < 2.5$ |
| Moderate Rain | $2.5 \leq i < 10.0$ |
| Heavy Rain | $10.0 \leq i < 50.0$ |
| Violent Rain | $i \geq 50.0$ |

i represents the daily precipitation.

## 3. Results

### 3.1. Accumulated Rainfall

Figure 3 depicts the spatial distributions of 96 h of accumulated rainfall during 12 July NCER, from UTC 00:00 on 10 July 2021 to UTC 23:00 on 13 July 2021. As indicated in Figure 3a, there were two heavy rainfall centers in 12 July NCER, one in the northwest of Shandong Province and the other in the eastern coastline region of Hebei Province. All the satellite-based QPE products show the similar rainfall region during this extreme rainstorm (Figure 3b–e), especially the IMERG products. As shown in Figure 3b,c, there is a significant overestimation of rainfall centers ($\geq$300 mm) and underestimation of other sporadic sub-heavy rainfall centers (northern Henan, southeastern Shanxi, northwestern Beijing, and southern Hebei) in IMERG products. Actual heavy rainfall centers can also be captured by GSMaP products, although there is an overestimation of actual heavy rainfall centers. As seen in Figure 3d,e, GSMaP products overestimate precipitation in the southeast of Hebei Province. It is discovered that the GSMaP products' overestimation region is dispersed in an arc form along the coast and near to the heavy rainfall center. The inability of GSMaP products to capture more dispersed sub-precipitation centers is a similar issue. GSMaP products continue to overestimate precipitation in the eastern coastline area of Shandong Province when compared to IMERG data in this sub-precipitation center. In general, IMERG products, particularly IMERG_ER, are superior at capturing heavy rainfall regions and centers than GSMaP products.

As presented in Figure 4, we can see the spatial distributions of 144 h of accumulated rainfall during 20 July NCER from UTC 00:00 on 17 July 2021 to UTC 23:00 on 22 July 2021. The 20 July NCER was the most severe and intense precipitation event in Northern China. The 20 July NCER heavy rainfall event had both long duration and high intensity, and it produced a variety of meteorological calamities in low-altitude ($\leq$1000 m) areas of central and northwestern Henan Province. As shown in Figure 4a–e, four satellite QPE products can capture a portion of the heavy rainfall center and heavy rainfall region.

However, IMERG products outperform GSMaP products in terms of capture accuracy over the rainfall center, particularly the IMERG_LR, which has an excellent fit to the real heavy rainfall center. Meanwhile, the GSMaP products vastly underestimated the intensity of this severe rainfall storm. It is interesting to note that the four QPE products in northwest Henan Province underestimated the amount of precipitation close to the Tai-hang Mountains (Figure 1). This is likely due to the fact that the topography of the study area, as well as the algorithms in QPE products, has an impact on the effectiveness of these QPE products in capturing heavy precipitation.

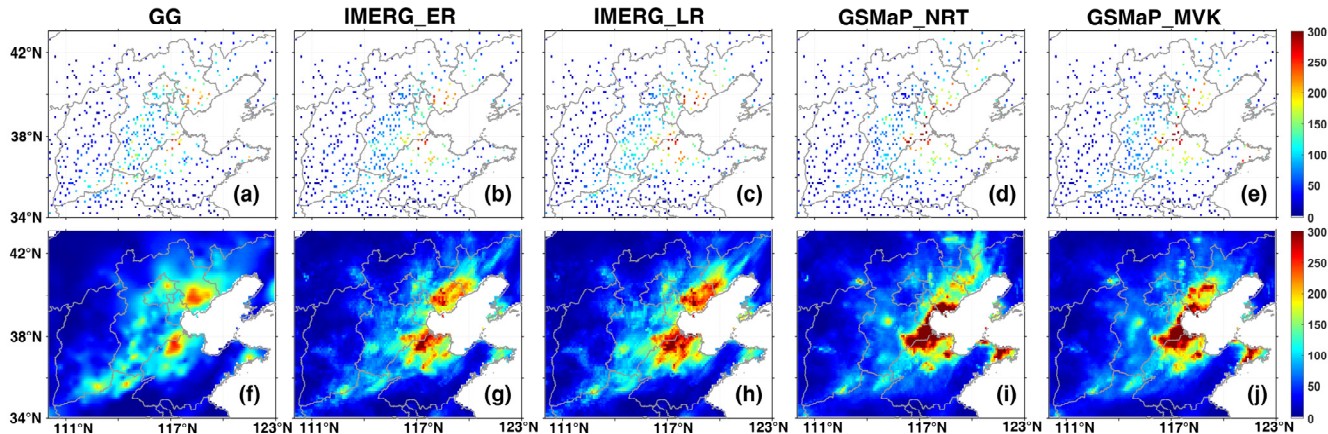

**Figure 3.** Spatial distribution of 96 h accumulated rainfall during 12 July NCER based on: (**a**–**e**) the grid data of GG, IMERG_ER, IMERG_LR, GSMaP_NRT and GSMaP_MVK; (**f**–**j**) the data of GGKRIG, IMERG_ER, IMERG_LR, GSMaP_NRT and GSMaP_MVK.

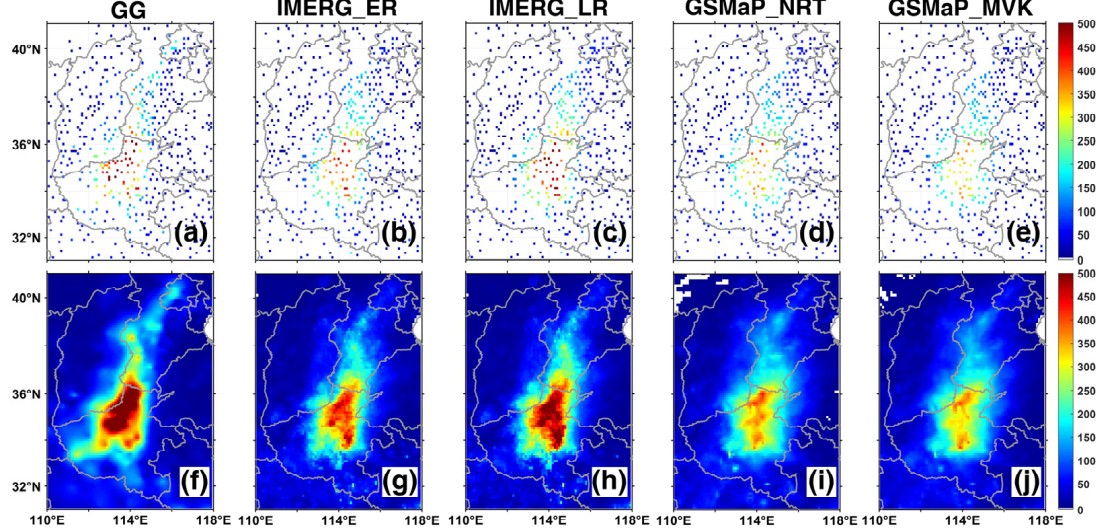

**Figure 4.** Spatial distribution of 144 h accumulated rainfall during 20 July NCER based on: (**a**–**e**) the grid data of GG, IMERG_ER, IMERG_LR, GSMaP_NRT and GSMaP_MVK; (**f**–**j**) the data of GGKRIG, IMERG_ER, IMERG_LR, GSMaP_NRT and GSMaP_MVK.

Figure 5 shows the spatial distributions of 108 h accumulated rainfall during the 5 October NCER from UTC 00:00 on 2 October 2021 to UTC 23:00 on 6 October 2021, excluding UTC 00:00 to 01:00 on 4 October 2021, UTC 03:00 on 4 October 2021, UTC 11:00 on 5 October 2021, UTC 13:00 to 14:00 on 5 October 2021, and UTC 10:00 to 15:00 on 6 October 2021. The 5 October NCER heavy rainfall center was situated in the west of central Shanxi Province and central of Shaanxi Province, and its heavy rainfall belt

stretches symmetrically from central Shanxi Province and Shaanxi Province to the north and south. However, Figure 5b,c show that IMERG products perform poorly in capturing the rainfall center in central Shanxi Province, which cannot be captured, whereas the major heavy rainfall center captured by IMERG products is in central Hebei Province. The IMERG products are considerably underestimated for estimating cumulative precipitation in heavy precipitation centers. This could be due to the complicated terrain of Shanxi Province, which has a basin in the center and hilly mountains to the east and west. The precipitation estimation of difficult terrain regions by IMERG products still has limitations, which caused it to estimate precipitation incorrectly and miss the Shanxi Province's heavy precipitation center. On the contrary, the GSMaP products agree very well with the actual precipitation, capturing the shape and location of the heavy rainfall belt, but have more intensive precipitation than the observations, especially over the central Shanxi province, Beijing, Tianjin, northeastern Hebei province, and northeastern Shandong province.

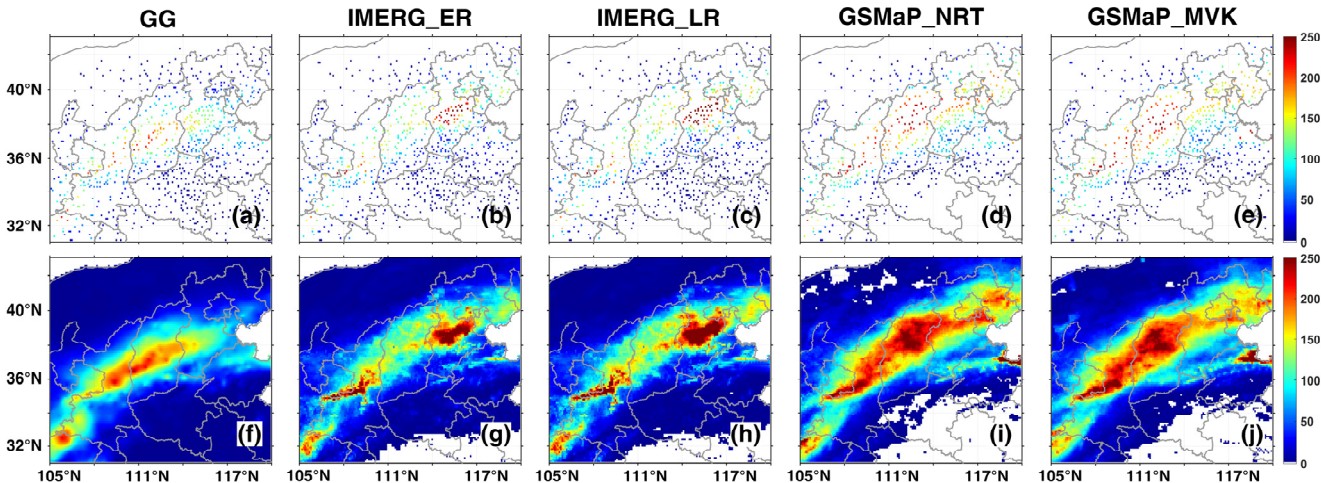

**Figure 5.** Spatial distribution of 108 h accumulated rainfall during 5 October NCER based on: (**a–e**) the grid data of GG, IMERG_ER, IMERG_LR, GSMaP_NRT and GSMaP_MVK; (**f–j**) the data of GGKRIG, IMERG_ER, IMERG_LR, GSMaP_NRT and GSMaP_MVK.

For quantitative assessment, Figure 6 depicts the scatter plots of IMERG and GSMaP satellite-based QPE products versus the GG in terms of accumulated precipitation during the three extreme rainstorms, with bulk statistics summarizing their performance displayed in each figure. On 12 July NCER (Figure 6a,d,g,j), IMERG_ER, IMERG_LR, GSMaP_NRT, and GSMaP_MVK had low CC (0.75, 0.75, 0.60, and 0.67) with GG, which indicates that IMERG_ER, IMERG_LR, GSMaP_NRT, and GSMaP_MVK failed to capture the spatial precipitation pattern. The index RB implies that IMERG products (IMERG_ER and IMERG_LR) and GSMaP products (GSMaP_NRT and GSMaP_MVK) overestimated the precipitation by about 3.16%, 11.75%, 14.73%, and 7.78%. As for RMSE and FSE, the IMERG_ER has the lowest RMSE (39.35 mm) and FSE (25.00) among the four products. The GSMaP products do not perform well in the 12 July NCER, especially GSMaP_NRT. GSMaP_NRT has the highest RMSE (60.46 mm), FSE (59.02), RB (14.73%), and the lowest CC (0.60). This may be related to the GSMaP_NRT products (Figure 3d), which overestimate the precipitation in the eastern coastal areas of the Beijing–Tianjin–Hebei Urban Agglomeration and the three regional boundaries of Hebei Province, Liaoning Province, and Inner Mongolia. As illustrated in Figure 6b,e,h,k, in the 20 July NCER, the four satellite-based QPE products underestimated precipitation to varying degrees; this occurred because the index RBs in IMERG products (IMERG_ER and IMERG_LR) and GSMaP products (GSMaP_NRT and GSMaP_MVK) are approximately −9.75%, −0.37%, −11.45%, and −17.08%. The CCs of the four satellite-based QPE products are all around 0.8 in 20 July NCER, with GSMaP MVK (0.82) performing best and IMERG ER (0.79) performing worst. In terms of RMSEs

and FSEs, the four satellite-based QPE products maintain high values (RMSE > 82.00 mm and FSE > 70.00), indicating that the overall accuracy of QPE products is poor with large errors in the 20 July NCER. This might be mainly because typhoon In-Fa and typhoon Cempaka did not directly affect the weather in the Northern China but instead transferred water vapor to Northern China through atmospheric circulation [9]. Satellite precipitation products were unable to correctly collect the required water vapor sources and infrared information, resulting in substantial deviation. On the 5 October NCER (Figure 6c,f,i,l), IMERG_ER, IMERG_LR, GSMaP_NRT, and GSMaP_MVK also had low CC (0.72, 0.70, 0.70, and 0.75) with GG, which indicates that the correlation between the satellite QPE products and the GG is not high in space and time. As for RBs, the four satellite-based QPE products exhibit varying degrees of overestimation, with the GSMaP_MVK (39.30%) being the most overestimated and the IMERG_ER (11.53%) being the least. In fact, the RB values of GSMaP products are generally high, which may be related to the overestimation of strong precipitation in northern Shanxi Province and the western part of Shaanxi Province by GSMaP products. During the 12 July NCER, the IMERG_ER had the lowest RMSE (45.29 mm) and FSE (36.49).

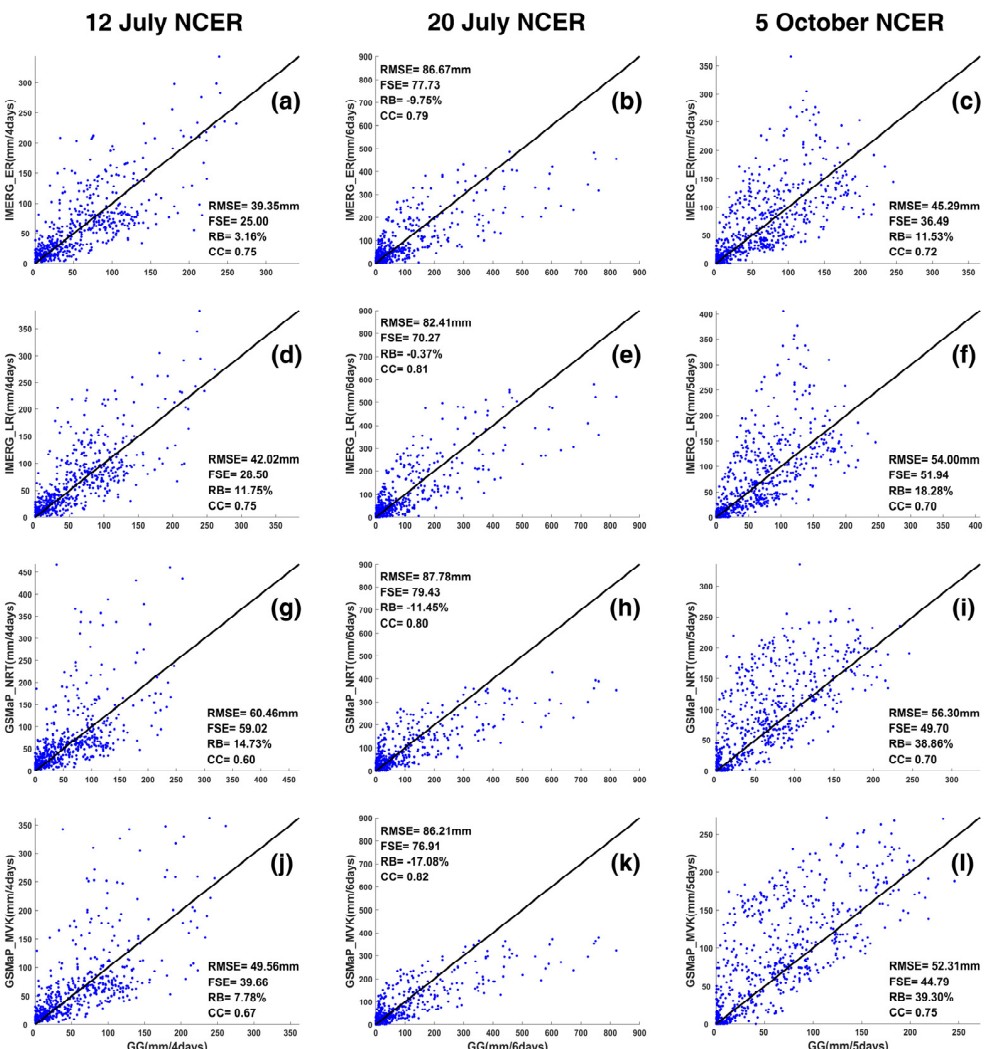

**Figure 6.** The scatter plots of GG versus (**a**–**c**) IMERG_ER; (**d**–**f**) IMERG_LR; (**g**–**i**) GSMaP_NRT; and (**j**–**l**) GSMaP_MVK for the three extreme rainstorms' accumulated precipitation shown in Figures 3–5.

The Taylor diagram (Figure 7) proposed by Taylor [70] is used to visualize the statistical summary of how well the four satellite-based QPE products agree with the GG in terms

of CC, RMSE, and standard deviation (SD) values to better evaluate the comprehensive performance of these four satellite-based QPE products. In the Taylor diagram, the product that performs best is the point that is closest to the reference data. As illustrated in Figure 7a,c, the IMERG_ER (the red fork) was the closest point to the GG (the black star), implying that the IMERG_ER had the best performance during the 12 July NCER and 5 October NCER. On the 20 July NCER (Figure 7b), the IMERG_LR outperformed the other three satellite-based QPE products because the red square is the closest point to the GG (the black star). In contrast, the GSMaP products show poorer performance in the three extreme rainstorms. Generally, these findings are consistent with the analysis of the scatter plots in Figure 6.

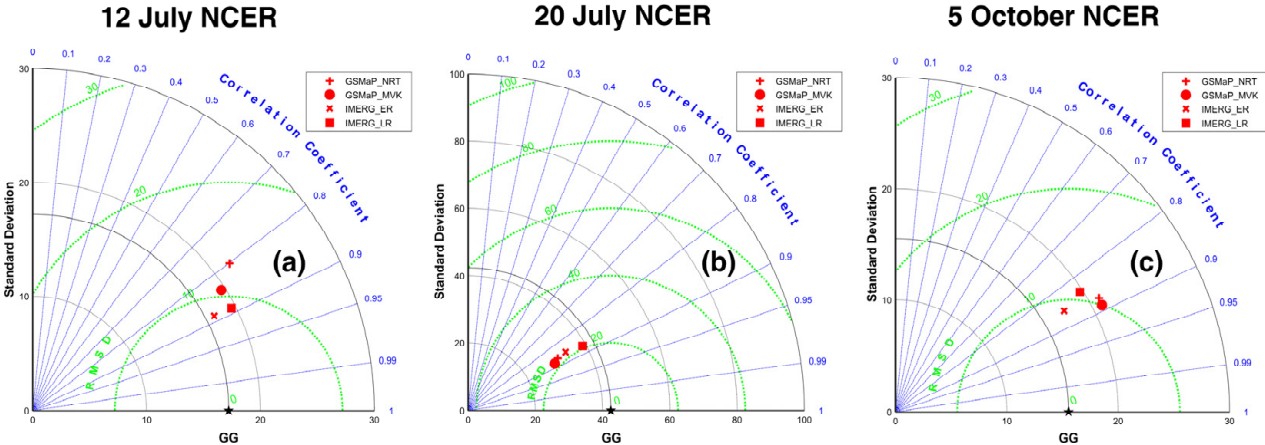

**Figure 7.** Taylor diagram showing the CC, SD and RMSE of accumulated rainfall between the GG and the satellite-based QPE products in three extreme rainstorms: (**a**) 12 July NCER; (**b**) 20 July NCER; and (**c**) 5 October NCER.

### 3.2. Mean Hourly Rainfall

The temporal variation of precipitation impacts the hydrologic cycle over the land surfaces where landslides, floods, and other hydro-related hazards could be triggered by rainfall storms. The mean hourly precipitation, specifically the average hourly precipitation in the study region, is utilized to examine the temporal variation features of precipitation. In order to better appreciate the performance of the satellite-based QPE products during the extreme rainstorms, the Figure 8 shows the time series of mean hourly precipitation for all rainfall products during the three extreme rainstorms, and the time series of the three extreme rainstorms are shown in Figure 8a (12 July NCER), Figure 8b (20 July NCER), and Figure 8c (5 October NCER), respectively.

As presented in Figure 8a, the 12 July NCER precipitation started on the night of 11 July with a major and minor peak. The major peak was observed at the 40th hour (UTC 15:00 on 11 July 2021) and the minor peak at the 85th hour (UTC 12:00 on 13 July 2021). It is noted that only the GSMaP_NRT has captured the major peaks, but the IMERG_ER failed to capture the minor peaks. In contrast, the GSMaP products significantly overestimated the two rainfall peaks. All the satellite-based QPE products slightly overestimated mean hourly precipitation to different degrees, with IMERG_ER (where the RB value is 0.04%) performing best. In terms of CCs, the general performance of IMERG products is superior to GSMaP products, and the CCs of IMERG products could approach 0.9, notably IMERG_LR (0.92). In general, IMERG products outperform GSMaP products in terms of mean hourly rainfall in all parameters.

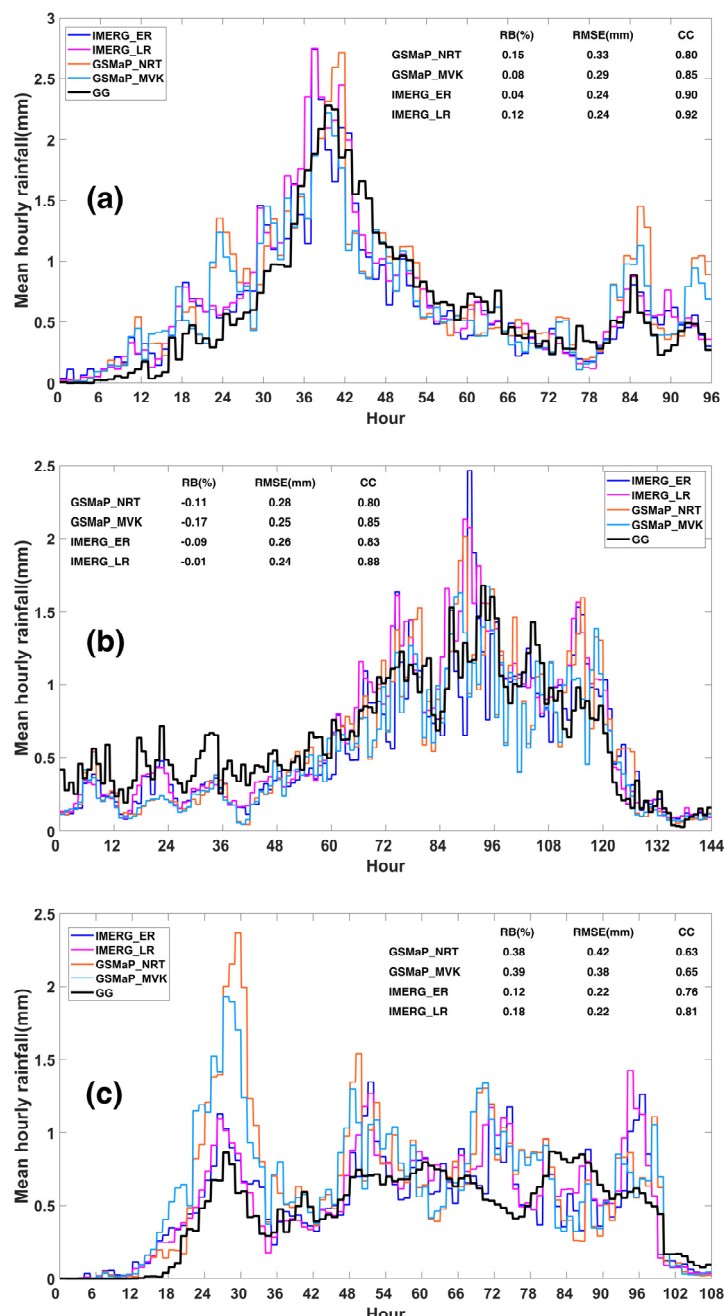

**Figure 8.** Time series of mean hourly precipitation of GG versus the satellite-based QPE products for the three extreme rainstorms: (**a**) 12 July NCER; (**b**) 20 July NCER; and (**c**) 5 October NCER.

Extreme precipitation events induced by typhoons, unlike other precipitation events, feature numerous peaks in the mean hourly precipitation, a substantial variability of precipitation, and a short time interval between each peak. Figure 8b shows that the highest peak occurred during the 94th hour (UTC 21:00 on 20 July 2021) and the heavy rainfall phase in the big peak area was concentrated in the period of UTC 00:00 on 20 July 2021 to UTC 23:00 on 21 July 2021. As shown in Figure 8b, no satellite-based QPE product can capture the highest precipitation peak. All satellite-based QPE products, with IMERG LR (the RB value is −0.01%) performing the best, slightly underestimated mean hourly precipitation to varying degrees. Evidently, the IMERG_LR performed well in capturing the changes of mean hourly rainfall, with the highest CC (0.88) and lowest RMSE (0.24 mm) and RB (−0.01%) among these four satellite-based QPE products.

As shown in Figure 8c, the fluctuation in precipitation during the 5 October NCER was flat and continuous. In the meantime, the highest peak was generated at the 82nd (UTC 15:00 on 5 October 2021). Evidently, the four satellite-based QPE products can capture the first peak (at 28th, UTC 03:00 on 3 October 2021) with overestimation, but they are unable to capture the highest peak. Additionally, in terms of RBs, the four satellite-based QPE products of 5 October NCER performed similarly to 12 July NCER, which was slightly overestimated precipitation. On the 5 October NCER, the IMERG_LR showed better agreement with the GG in the highest CC (0.81), lower RB (0.18%), and RMSE (0.22 mm) values. Additionally, the RBs and RMSEs of GSMaP products were almost twice as high as those of IMERG, demonstrating that the IMERG products were more accurate overall and had fewer mistakes than the products of GSMaP in the 5 October NCER.

Generally, it is noted that several interesting phenomena are shown in Figure 8: (1) all satellite-based QPE products overestimate precipitation when the precipitation is light and underestimate it when the precipitation is heavy; (2) satellite-based QPE products have a peak that is 2–4 h earlier than the GG peak, which may aid in forecasting actual heavy precipitation.

### 3.3. Contingency Information

POD, CSI, and FAR contingency scores are commonly used to evaluate the performance of satellite-based QPE products that anticipate varied rainfall thresholds [71]. According to the Chinese meteorological standard about the grade of rainfall in short-term weather service [72], the hourly rainfall can be divided into six grades: short-term light rain (rainfall < 2 mm/h), short-term moderate rain (2 mm/h ≤ rainfall ≤ 3.9 mm/h), short-term heavy rain (4 mm/h ≤ rainfall ≤ 7.9 mm/h), short-term rainstorm (8 mm/h ≤ rainfall ≤ 19.9 mm/h), short-term extreme rainstorm (20 mm/h ≤ rainfall ≤ 50 mm/h) and short-term extraordinary rainstorm (50 mm/h > rainfall). These criteria can be used to assess the performance of satellite-based QPE products in monitoring and issuing warnings about the weather during a relatively short period of time.

As shown in Figure 9a–c, the IMERG products in three extreme rainstorms perform better with high POD (>30%) in short-term light rain. However, as for the capture of short-term rainstorms to short-term extreme rainstorms, the GSMaP products outperform the IMERG products. This phenomenon suggests that GSMaP products have the potential to catch short-term rainstorms and short-term extreme rainstorm events. It is interesting that the four satellite-based QPE products have greater POD values in every grade of rainfall generated by typhoons (Figure 9b) than those caused by non-typhoons (Figure 9a,c). It is worthy of note that Figure 9g,i exhibit similar FARs trends, showing that the false alarm degree of GSMaP products for assessing short-term rainstorms and short-term extreme rainstorms is better than IMERG products. However, it can be observed from Figure 9h that IMERG products perform better than GSMaP products. In order to pinpoint precise precipitation occurrences, CSI thoroughly evaluates POD and FAR, which determines the genuine quantity of satellite-based QPE products to detect real precipitation events. Interestingly, in terms of the CSIs on 12 July NCER (Figure 9d) and 5 October NCER (Figure 9f), with the grades of short-term rainstorm and short-term extreme rainstorm, the GSMaP products also outperform IMERG products, which is consistent with their performance in FARs. Also, on 20 July NCER, the general performance of IMERG products was superior to that of GSMaP products regarding CSIs.

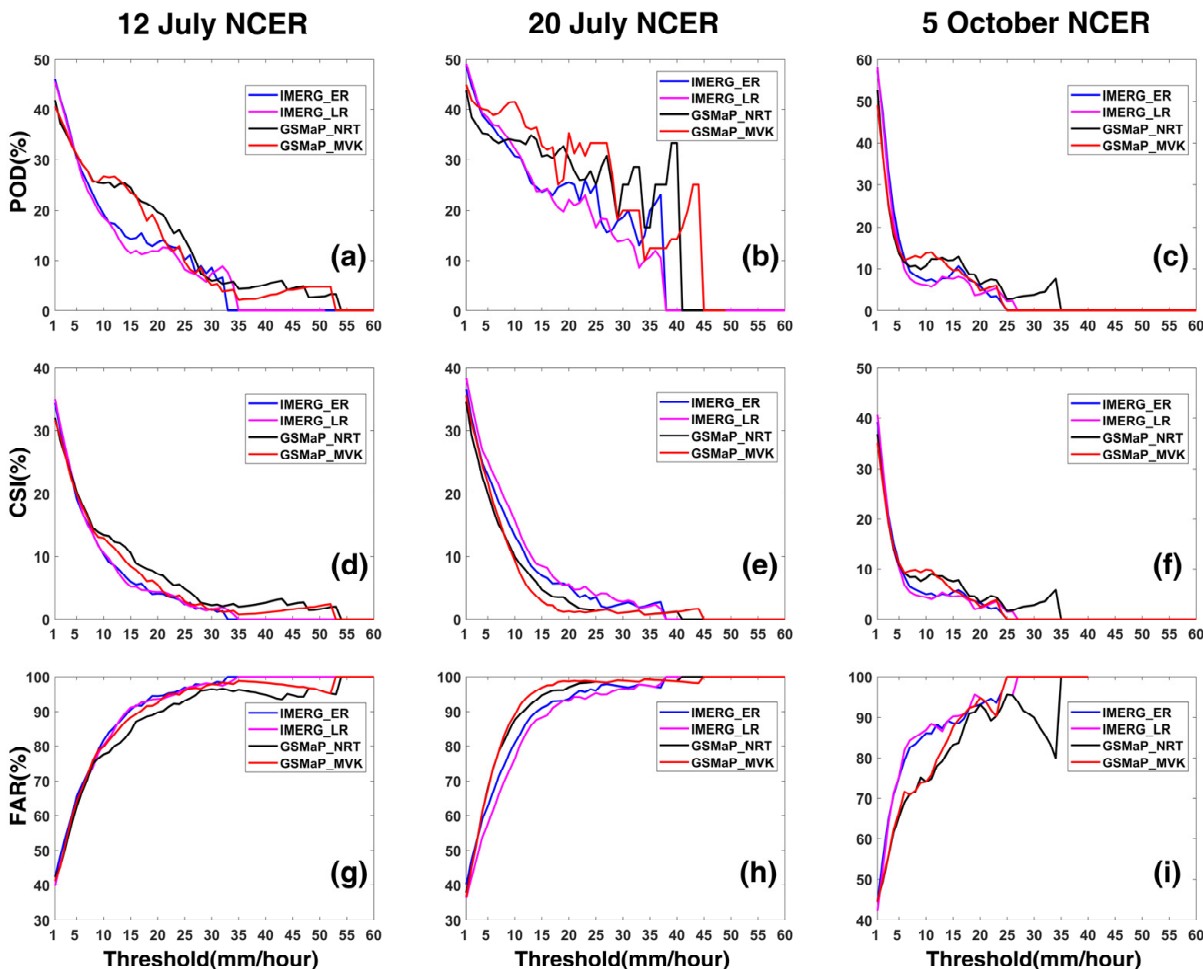

**Figure 9.** Contingency statistics computed from IMETG and GSMaP hourly precipitation during three extreme rainstorms: (**a**–**c**) Probability of Detection (POD); (**d**–**f**) Critical Success Index (CSI); and (**g**–**i**) False Alarm Ration (FAR).

Overall, it can be believed that the GSMaP products may have the capacity to detect the short-term rainstorm events during the rainstorms that are brought by non-typhoons.

## 4. Discussion

Different satellite-based QPE products provide different benefits in terms of monitoring precipitation occurrences. At the same time, it is extremely important to summarize the spatial and temporal variation of precipitation forecast by satellite-based QPE products for meteorological forecast. In this study, the ground rain gauge observations are used as a reference to assess the performance of IMERG (IMERG_ER & IMERG_LR) and GSMaP (GSMaP_NRT & GSMaP_MVK) satellite-based QPE products during the 12 July NCER, 20 July NCER, and 5 October NCER in 2021. The accumulated rainfall, mean hourly rainfall, and the contingency scores of four satellite-based QPE products were computed for assessment. The spatial distribution and time series of precipitation during the three extreme rainstorms for various satellite-based QPE products, as well as their ability to detect precipitation, were analyzed.

As shown in Figures 3 and 5, it can be found that both IMERG products and GSMaP products overestimated precipitation in low-altitude (≤1000 m) areas during the 12 July NCER and 5 October NCER, which were extreme precipitation occurrences induced by non-typhoons. As indicated in Figures 4 and 5, the 20 July NCER had heavy rainfall centers in the north and northwest of Henan Province, and the 5 October NCER showed heavy

rainfall centers in the central and western Shanxi Province and central Shaanxi Province. In addition, the GSMaP products and IMERG products failed to capture heavy precipitation centers and magnitudes during the 20 July NCER (Figure 4) and 5 October NCER (Figure 5) precipitation processes. Satellite precipitation sensors mainly include IR, as well as passive and active microwave sensors. Both IMERG and GSMaP precipitation products integrate passive microwave radiometer data with infrared radiometer data and indirectly estimate precipitation on the ground by detecting cloud top temperature information using infrared sensors [25,62]. The main theoretical basis of IR precipitation estimation is that when the cloud top temperature in a cloud area is below a certain threshold and the extent of the region continues to expand, or the temperature gradient between the cloud top core area and the surrounding cloud area is large, or when the temperature of the cloud area has a downward trend, it indicates the possibility of further development of strong convection and will produce precipitation [23]. There is a limitation in using IR to estimate precipitation, whereby the cloud top motion may not match the precipitation movement [73,74], and this limitation may be one of the reasons why the four satellite-based QPE products do not capture the center of actual heavy precipitation well.

However, retrieval algorithms, sensors, geography, and altitude may all have an impact on the accuracy of QPE products [27]. Previous research has shown that the estimation of precipitation from QPE products is highly dependent on elevation and topography [27,29,73]. In 20 July NCER, the primary heavy rainfall centers were located at the foot of the Tai-hang Mountains in the northwest of Henan Province, whereas QPE products failed to capture heavy rainfall near the Tai-hang Mountains. Similarly, Shanxi province, the site of the 5 October NCER, has complex terrain; its east and west sides are mountainous and hilly uplifts, and the central part is a string of beaded basin subsidence, while plains are distributed between them. The IMERG and GSMaP satellite precipitation estimate algorithms struggle in this type of terrain. Simultaneously, altitude is a significant element influencing the accuracy of QPE products. In Figure 3 through Figure 5, we can observe that QPE products considerably underestimate precipitation in high-altitude sporadic sub-heavy rainfall centers, such as the sub-heavy rainfall centers in northern Henan, southeastern Shanxi Province, northwestern Beijing, and southern Hebei Province that occurred in the 12 July NCER process. Interestingly, it can be found in the 5 October NCER (Figure 5) that the IMERG's heavy rainfall centers were concentrated over low-altitude areas ($\leq$1000 m), while GSMaP's heavy rainfall centers were concentrated in high-altitude areas (>1000 m) and the high-elevation areas received more precipitation than the low-elevation areas. It is possible that the algorithm of GSMaP misclassifies the cold surface on the top of the mountain as rain clouds, resulting in an overestimation of rainfall. As shown in Figure 6, satellite-based QPE products show fair performance (the CC is low, and the RMSE is large). This may be due to the limited spatial resolution of satellite precipitation products ($0.1° \times 0.1°$), since precipitation within an area may occur at a scale lower than the satellite pixel size [75]. The IMERG products were inferior to the GSMaP satellite precipitation products in capturing precipitation centers. Considering the quantitative evaluation (Figure 6) and Taylor diagram analysis (Figure 7), it can be found that the comprehensive performance of IMERG products was better than that of GSMaP satellite products in the three extreme rainstorms, and the IMERG_ER was the best in 12 July NCER and 5 October NCER, while the IMERG_LR was the best in 20 July NCER.

Figure 8 displays the time series of mean hourly precipitation for the three extreme rainstorms. Another major aspect impacting the accuracy of satellite-based QPE products estimates is precipitation intensity [30,31]. The satellite-based QPE products have been proven to overestimate light rain and underestimate heavy rain in previous studies [20,30–32]. It is noted that the four satellite-based QPE products have the phenomenon of predicting the peak precipitation in advance (such as the highest precipitation peaks of the three extreme rainstorms). It can be found that all the satellite-based QPE products are inclined to overestimate precipitation amounts when the precipitation amount is small, while they are going to underestimate precipitation amounts when the precipitation

amount are large. In view of this scenario of satellite-based QPE products, we plotted the hourly spatial distribution maps of three extreme precipitation events. According to the spatial distribution maps, satellites typically fail to detect or underestimate precipitation in places with high elevations and complicated terrain, and even overestimate precipitation in low-altitude ($\leq$1000 m) and flat areas. This demonstrates that altitude and terrain complexity continue to be major variables influencing the accuracy of satellite-based QPE products. As for the performance of mean hourly precipitation, GSMaP products and IMERG products estimated the peak of precipitation 2–4 h in advance, but there was a significant overestimation of precipitation intensity, which is consistent with the findings by Chen et al. [20]. Interestingly, as shown in Figure 8, IMERG_ER lagged behind IMERG_LR in estimating precipitation, which could be due to the missed precipitation event caused by the forward-morphing algorithm used by IMERG_ER, or it could be due to the fact that IMERG_ER uses more IR precipitation estimates than IMERG_LR.

As shown in Figure 9, the GSMaP products may have the capacity to detect the short-term rainstorm events during the three rainstorms. This is consistent with Saber's previous research [15], which found that although GSMaP_NRT and GSMaP_MVK can accurately identify rainfall events, their precipitation amounts are not precise enough. According to the PODs, CSIs and FARs, GSMaP products perform more effectively in non-typhoon extreme precipitation events (12 July NCER and 5 October NCER), whereas IMERG products do well in typhoon extreme precipitation events (20 July NCER). At the same time, there are several differences between GSMaP_NRT and GSMaP_MVK (or IMERG_ER and IMERG_LR), which may be attributable to the product having various input sources and retrieval algorithms.

Up to date, to our knowledge, the current study is also limited in the distribution density of surface rainfall stations, and the results and discussions in this thesis are based on the results of ground rainfall station data to evaluate the satellite-based QPE products, which implies that there are some limitations in this evaluation method itself. The distribution density of ground rainfall stations is also one of the factors that affects the performance of satellite-based QPE products [15], because the topography of Northern China is complex and the density of ground rainfall stations is unevenly distributed. In the sparse distribution of rainfall stations, the performance of satellite-based QPE products is not necessarily its actual performance, which may lead to unsatisfactory evaluation results of satellite-based QPE products. Although we want to evaluate the performance of satellite-based QPE products more accurately due to the current inability to obtain regional rain gauge observations in various provinces, we can only use national station rainfall data for research. However, scholars interested in satellite-based QPE product evaluation can consider using the regional rain gauge observations for research. At the same time, we feel that the current national station rainfall stations are mostly located in flat terrain and low-altitude ($\leq$1000 m) locations, making it impossible to assess the effectiveness of satellite-based QPE products in high-altitude mountainous areas, particularly for snowfall event monitoring.

## 5. Conclusions

This study evaluates the performance of several mainstream satellite-based QPE products (IMERG_ER, IMERG_LR, GSMaP_NRT and GSMaP_MVK) for three extreme precipitation events in Northern China, using numbers of rain gauge observations as references. The evaluation metrics, including RB, RMSE, CC, FSE, POD, CSI, and FAR, were applied to examine the performance of satellite-based QPE products in terms of accumulated rainfall, time series rainfall, and contingency. The important findings are as follows:

(1)　Spatially, IMERG products outperform GSMaP products because the strong precipitation centers captured by IMERG products are highly consistent with the observations, especially in extreme precipitation events in areas with relatively flat terrain and low-altitude ($\leq$1000 m). When compared to the GSMaP products during the three

extreme rainstorms, the IMERG products consistently had greater CCs, lower RBs, and RMSEs. Also, IMERG products are better at capturing the spatial distribution of precipitation.

(2) Temporally, GSMaP satellite-based QPE products capture most precipitation peaks but significantly overestimate actual precipitation intensity, whereas IMERG satellite-based QPE products have a better fit with actual precipitation. At the same time, four satellite-based QPE products underestimate (overestimate) precipitation when the actual precipitation is heavy (light). In the comprehensive analysis, the IMERG_LR performs best, as it has the highest CCs (0.92, 0.88 and 0.81), lower RBs (0.12%, −0.01% and 0.18%) and RMSEs (0.24 mm, 0.24 mm, and 0.22 mm), whether in typhoon or non-typhoon extreme precipitation events.

(3) In terms of POD, CSI, and FAR performance in extreme precipitation events, the GSMaP products perform more effectively in non-typhoon extreme precipitation events (12 July NCER and 5 October NCER), whereas IMERG products behave well in typhoon extreme precipitation events (20 July NCER). GSMaP satellite-based QPE products outperform IMERG satellite-based QPE products in estimating the precipitation of short-term rainstorm events.

In general, IMERG satellite-based QPE products could be given priority in the application of precipitation prediction because they outperform GSMaP satellite-based QPE products in predicting heavy precipitation, but for short-term heavy rainfall forecasts, GSMaP products can be good alternatives.

This study assesses the accuracy of the satellite-based QPE products (IMERG_ER, IMERG_LR, GSMaP_NRT, and GSMaP_MVK) during the three extreme rainstorms over Northern China in 2021, using the hourly ground-based rain gauge observations as reference. The gauge observations are compared with the IMERG and GSMaP products for grids that were overlapped with at least one gauge. According to this comparative study, QPE products generally overestimate the heavy rainfall caused by non-typhoons and underestimate the heavy rainfall caused by typhoons. Precipitation intensity, sensors, geography, and other factors can impact the accuracy of satellite-based QPE products, resulting in varying degrees of accuracy in different regions. This indicates that under extreme precipitation circumstances, satellite-based QPE products must be corrected for inaccuracies in varied terrains, sensors, and precipitation intensities. Nowadays, some studies [13] have found that the distribution density of ground rainfall stations is also one of the important factors affecting the satellite precipitation assessment.

In conclusion, we propose that in the future, more ground rainfall stations be used to record actual precipitation data in areas with high spatial heterogeneity and complex terrain in order to evaluate the performance of satellite-based QPE products and avoid the uncertainty in the potential assessment caused by the uncertainty of the reference data. Furthermore, satellite-based QPE products employ IR to measure precipitation, and there may be some mismatch between the cloud top motion and the actual precipitation movement, which may contribute to precipitation misjudgment. We propose that data programmers enhance the retrieval method of satellite-based QPE products for computing weights and make full use of IR-based information. In order to increase the accuracy of satellite-based QPE products, product developers must thoroughly account for the effects of precipitation intensity, sensors, location, and other elements during product development. In addition, the weather forecasters must modify the precipitation prediction in the actual forecast application in view of these variables. At the same time, the research of satellite-based QPE products in the GPM era should be emphasized in the current climate background to improve their accuracy and enhance their applicability in complex terrain areas to better serve weather forecasting and monitoring.

**Author Contributions:** S.C. conceived and designed the experiments; H.Z. performed the experiments and analyzed the data; Z.L. prepared the data and proofread the paper; X.L. and L.G. helped analysis the results; H.Z. and S.C. wrote the paper. All authors have read and agreed to the published version of the manuscript.

**Funding:** This research was partially sponsored by the National Natural Science Foundation of China (41875182) and Key Laboratory of Environment Change and Resources Use in Beibu Gulf (NNNU-KLOP-K2103) at Nanning Normal University, Guangxi Key R&D Program (Grant No. AB22035016, 2021AB40137), and The High-level Talent Program (E2290702) in the Northwest Institute of Eco-Environment and Resources, Chinese Academy of Sciences.

**Informed Consent Statement:** Not applicable.

**Data Availability Statement:** Data available in a publicly accessible repository that does not issue DOIs. Publicly available datasets were analyzed in this study. The rain gauge observations data can be found in China Meteorological Data Service Center (http://data.cma.cn/, accessed on 7 October 2021). Additionally, the IMERG data can be obtained in NASA (https://gpm.nasa.gov/, accessed on 25 October 2021) and the GSMaP data can be available in JAXA (http://www.gportal.jaxa.jp/, accessed on 25 October 2021).

**Acknowledgments:** We thank all organizations for providing the IMERG and GSMaP as well as the rain gauge observations data freely to the public. We also highly appreciate the detailed reviews and the helpful comments and suggestions from the five reviewers.

**Conflicts of Interest:** The authors declare no conflict of interest.

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
