# Peer review of "Comparison of Satellite Precipitation Products: IMERG and GSMaP with Rain Gauge Observations in Northern China"

_remotesensing, doi:10.3390/rs14194748_

Round 1
Reviewer 1 Report (Previous Reviewer 1)
REVIEW of the manuscript "Evaluation and Uncertainty Analysis of IMERG and GSMaP Satellite Precipitation Products in Northern China" by Huiqin Zhu, Sheng Chen, Zhi Li, Liang Gao, Xiaoyu Li [Title. Remote Sens. 2022,14, x. https://doi.org/10.3390/xxxxx].
This manuscripts evaluates the performance of several mainstream satellite-based QPE products (IMERG_ER, IMERG_LR, GSMaP_NRT and GSMaP_MVK) for three extreme precipitation events in Northern China, using numbers of rain gauge observations as references. The evaluation metrics, including RB, RMSE, CC, FSE, POD, CSI and FAR, were applied to examine the performance of satellite-based QPE products in terms of accumulated rainfall, time series rainfall and contingency. The given results are promising in Northern China and wider. This version of the manuscript is actually an improved version already submitted to this journal for review. The authors have made a great effort to improve the quality of the manuscript.
However, I think the manuscript can be even better if it is documented with more references outside of China with a critical review and discussion. The authors should emphasize even more the novelty of their research as well as the advantages of the presented method considering the rank of the journal in which they want to publish their manuscript.
Author Response
Response to Reviewer 1 Comments
First of all, we are grateful to your helpful suggestions. We accepted all of your comments and substantially revised the text. Below, we repeat your comments first and then give our point-by-point response. All responses are in red font for clarity of reading.
Point 1: This manuscripts evaluates the performance of several mainstream satellite-based QPE products (IMERG_ER, IMERG_LR, GSMaP_NRT and GSMaP_MVK) for three extreme precipitation events in Northern China, using numbers of rain gauge observations as references. The evaluation metrics, including RB, RMSE, CC, FSE, POD, CSI and FAR, were applied to examine the performance of satellite-based QPE products in terms of accumulated rainfall, time series rainfall and contingency. The given results are promising in Northern China and wider. This version of the manuscript is actually an improved version already submitted to this journal for review. The authors have made a great effort to improve the quality of the manuscript. However, I think the manuscript can be even better if it is documented with more references outside of China with a critical review and discussion. The authors should emphasize even more the novelty of their research as well as the advantages of the presented method considering the rank of the journal in which they want to publish their manuscript.
Response 1: Thanks for your good comments. We have gathered and referenced additional international literature on satellite-based QPE products and have updated and changed them where necessary, such as the Abstract (line 25-27 and 34–39) and Introduction (line 45-46 and 116–123). At the same time, we emphasize even more the novelty of paper, the details in the Introduction (line 133–136) and Conclusion (line 664–671).
- The details in lines of 25-27 and 34-39 are as follows:
We examined the spatial distribution of cumulative precipitation and the temporal distribution of hourly average precipitation for three severe precipitation occurrences using these assessment metrics.
The accuracy of satellite-based QPE products may be influenced by precipitation intensity, sensors, terrain, and other variables. Therefore, in accordance with our recommendations, more ground rainfall stations should be used to collect actual precipitation data in regions with high levels of spatial heterogeneity and complex topography, and data programmers should strengthen the weights computation retrieval technique and fully utilize infrared (IR)-based data.
- The details in lines of 45-46 and 116–123 are as follows:
Under a changing climate, both the frequency and intensity of precipitation events tend to increase in many regions[1,2].
As far as we know, recent studies overseas generally focus on the long-term series assessment of satellite-based QPE products[36–40], and few studies have examined the performance of satellite-based QPE products in short-term emergencies. At the moment, many scholars are primarily interested in evaluating the temporal and spatial distribution characteristics of long-term series of meteorological events such as drought and precipitation using satellite-based QPE products at daily scale data or monthly scale data as basic research data [41–44].
- The details in the Introduction (line 133–136) and Conclusion (line 664–671) are as follows:
There are few reports on the accuracy of V06 IMERG and V07 GSMaP in the range of inland regions in Northern China, especially for comparing the extreme precipitation events not originating from typhoons with those originating from typhoons.
Furthermore, satellite-based QPE products employ IR to measure precipitation, and there may be some mismatch between the cloud top motion and the actual precipitation movement, which may contribute to precipitation misjudgment. We propose that data programmers enhance the retrieval method of satellite-based QPE products for computing weights and make full use of IR-based information. In order to increase the accuracy of satellite-based QPE products, product developers must thoroughly account for the effects of precipitation intensity, sensors, location, and other elements during product development.
- The new references we added are as follow:
Reference:
- Du, H.; Alexander, L.V.; Donat, M.G.; Lippmann, T.; Srivastava, A.; Salinger, J.; Kruger, A.; Choi, G.; He, H.S.; Fujibe, F.; et al. Precipitation From Persistent Extremes Is Increasing in Most Regions and Globally. Ge-ophys. Res. Lett. 2019, 46, 6041–6049, doi:10.1029/2019GL081898.
- Pfahl, S.; O’Gorman, P.A.; Fischer, E.M. Understanding the Regional Pattern of Projected Future Changes in Extreme Precipitation. Nature Clim Change 2017, 7, 423–427, doi:10.1038/nclimate3287.
- Aslami, F.; Ghorbani, A.; Sobhani, B.; Esmali, A. Comprehensive Comparison of Daily IMERG and GSMaP Satellite Precipitation Products in Ardabil Provin. 16.
- Nascimento, J.G.; Althoff, D.; Bazame, H.C.; Neale, C.M.U.; Duarte, S.N.; Ruhoff, A.L.; Gonçalves, I.Z. Evaluating the Latest IMERG Products in a Subtropical Climate: The Case of Paraná State, Brazil. 2021, 18.
- Sharma, S.; Chen, Y.; Zhou, X.; Yang, K.; Li, X.; Niu, X.; Hu, X.; Khadka, N. Evaluation of GPM-Era Satellite Precipitation Products on the Southern Slopes of the Central Himalayas Against Rain Gauge Data. Remote Sensing 2020, 12, 1836, doi:10.3390/rs12111836.
- Nwachukwu, P.N.; Satge, F.; Yacoubi, S.E.; Pinel, S.; Bonnet, M.-P. From TRMM to GPM: How Reliable Are Satellite-Based Precipitation Data across Nigeria? Remote Sensing 2020, 12, 3964, doi:10.3390/rs12233964.
- Sharma, S.; Khadka, N.; Hamal, K.; Shrestha, D.; Talchabhadel, R.; Chen, Y. How Accurately Can Satellite Products (TMPA and IMERG) Detect Precipitation Patterns, Extremities, and Drought Across the Nepalese Himalaya? Earth and Space Science 2020, 7, doi:10.1029/2020EA001315.
- Darand, M.; Siavashi, Z. An Evaluation of Global Satellite Mapping of Precipitation (GSMaP) Datasets over Iran. Meteorol Atmos Phys 2021, 133, 911–923, doi:10.1007/s00703-021-00789-y.

Reviewer 2 Report (Previous Reviewer 2)
Mayor comments
Title: Too general, you are just evaluating the products for some case studies under specific conditions.
Abstract: Overall the abstract does not summary all the important information that it should include. What is the methodology used? Authors just mention the data used, a bunch of metrics and focus on the results. It is not enough, it is missing relevant important as beyond the fact that it is useful for developers, why this research is relevant for science?
Introduction: More that a bunch of studies describing extreme precipitation events. What are the dynamics and physics associated to this events? How climate change has modified the tendencies over precipitation patters? What are the main research questions? More that a manuscripts just evaluating precipitation products.
Datasets: You should also mention the natural shortcoming and uncertainties for each satellite products.
Conclusion: Even if the amount of rain gauge increases over those region, how this will improve the performance of the satellite products?
Typos and minor comments
L50 “cli-mate”
L23-24 This sentence does not make sense in the context of the Abstract “The grids contain at least one gauge for both gauge observations and the satellite-based quantitative precipitation estimation (QPE) products.
L39 Any reference for this affirmation?, which regions?
L138 What do you mean by “deeply”?
L223. Could you provide some references where this geostatistical method (kriging) was used for precipitation?
Author Response
Response to Reviewer 2 Comments
First of all, we are grateful to your helpful suggestions. We accepted all of your comments and substantially revised the text. Below, we repeat each your comments first and then give our point-by-point response. All responses are in red font for clarity of reading.
Point 1: Title: Too general, you are just evaluating the products for some case studies under specific conditions.
Response 1: Thanks. We have revised the title of the article to “Comparison of Satellite Precipitation Products: IMERG and GSMaP with Rain Gauge Observations in Northern China” in the revised version (line 2-3).
Point 2: Abstract: Overall the abstract does not summary all the important information that it should include. What is the methodology used? Authors just mention the data used, a bunch of metrics and focus on the results. It is not enough, it is missing relevant important as beyond the fact that it is useful for developers, why this research is relevant for science?
Response 2: Thanks. We have revised our Abstract based on your comments. See lines 25-27 and 34-41 for details.
(1) The details in lines of 25-27:
We examined the spatial distribution of cumulative precipitation and the temporal distribution of hourly average precipitation for three severe precipitation occurrences using these assessment metrics.
(2) The details in lines of 34-41:
The accuracy of satellite-based QPE products may be influenced by precipitation intensity, sensors, terrain, and other variables. Therefore, in accordance with our recommendations, more ground rainfall stations should be used to collect actual precipitation data in regions with high levels of spatial heterogeneity and complex topography, and data programmers should strengthen the weights computation retrieval technique and fully utilize infrared (IR)-based data. Furthermore, this study is expected to give helpful feedback to the algorithm developers of IMERG and GSMaP products as well as those researchers for the use of IMERG and GSMaP products in applications.
Point 3: Introduction: More that a bunch of studies describing extreme precipitation events. What are the dynamics and physics associated to this events? How climate change has modified the tendencies over precipitation patters? What are the main research questions? More that a manuscripts just evaluating precipitation products?
Response 3: Thanks. Our research focuses on examining the uncertainty of satellite-based QPE products, understanding the origins of the uncertainty, and ultimately giving suggestions for enhancing satellite precipitation data to improve weather proximity forecasting. As a result, we refer to papers on satellite-based QPE products precipitation assessment more often. At the same time, in response to your recommendation, we also refer to some of the literature on climate descriptions, as seen line 45–46 and 57-59.
- The details in lines of 45-46:
Under a changing climate, both the frequency and intensity of precipitation events tend to increase in many regions [1,2].
- The details in lines of 57-59:
Northern China, especially in Henan Province, mainly affected by typhoon In-Fa and typhoon Cempaka [8,9], has an extreme precipitation event on July 20, 2021.
(3) The new references we added are as follow:
Reference:
- Du, H.; Alexander, L.V.; Donat, M.G.; Lippmann, T.; Srivastava, A.; Salinger, J.; Kruger, A.; Choi, G.; He, H.S.; Fujibe, F.; et al. Precipitation From Persistent Extremes Is Increasing in Most Regions and Globally. Ge-ophys. Res. Lett. 2019, 46, 6041–6049, doi:10.1029/2019GL081898.
- Pfahl, S.; O’Gorman, P.A.; Fischer, E.M. Understanding the Regional Pattern of Projected Future Changes in Extreme Precipitation. Nature Clim Change 2017, 7, 423–427, doi:10.1038/nclimate3287.
- Xu, H.; Duan, Y.; Li, Y.; Wang, H. Indirect Effects of Binary Typhoons on an Extreme Rainfall Event in Henan Province, China From 19 to 21 July 2021: 2. Numerical Study. JGR Atmospheres 2022, 127, doi:10.1029/2021JD036083.
- Nie, Y.; Sun, J. Moisture Sources and Transport for Extreme Precipitation Over Henan in July 2021. Geophysical Research Letters 2022, 49, doi:10.1029/2021GL097446.
Point 4: Datasets: You should also mention the natural shortcoming and uncertainties for each satellite products.
Response 4: Thank you for your helpful comments. In this section, we just give a brief overview of the basic information about the data used. As for the uncertainties and shortcomings of these products are the focus of our work in this paper, we illustrate the uncertainties and shortcomings of the products through some columns of evaluation metrics, the details of which can be read in the discussion section. See lines of 525-536 and 551-560 for details. And we give a table to summarize the major similarities and differences among the four datasets used in the current manuscript. See lines 228-233 for details.
- The details in lines of 228-233:
The primary differences and similarities characteristic of satellite-based QPE products that are utilized in this article are summarized in Table 1.
|
Product |
Spatial / Temporal Resolution |
Spatial Domain |
Main Input Data |
Latency |
Applications |
|
IMERG_ER |
0.1°/ 0.5 hour |
90°N-90°S |
PMW, DPR1, GMI2, PR3, TMI4, AMSR25, SSMIS6 |
4 hours |
Prediction of flash floods and precipitation |
|
IMERG_LR |
0.1°/ 0.5 hour |
90°N-90°S |
PMW, DPR, GMI, PR, TMI, AMSR2, SSMIS |
14 hours |
Water resource management |
|
GSMaP_NRT |
0.1°/ 1 hour |
60°N-60°S |
TMI, GMI, SSMIS, PR, DPR, AMSU-A7/ AMSU-B8 |
4 hours |
Real-time applications |
|
GSMaP_MVK |
0.1°/ 1 hour |
60°N-60°S |
TMI, GMI, SSMIS, PR, DPR, AMSU-A/ AMSU-B |
3 days |
Water resource management |
Table 1. The major differences and common features of satellite-based QPE products.
1 GPM Dual-frequency Precipitation Radar; 2 GPM Microwave Imager; 3 TRMM Precipitation Radar; 4 TRMM Microwave Imager; 5 Advanced Microwave Scanning Radiometer Version 2; 6 Special Sensor Microwave Imager/Sounder; 7 Advanced Microwave Sounding Unit A; 8 Advanced Microwave Sounding Unit B.
- The details in lines of 525-536:
Both IMERG and GSMaP precipitation products integrate passive microwave radiometer data with infrared radiometer data and indirectly estimate precipitation on the ground by detecting cloud top temperature information using infrared sensors [25,62]. The main theoretical basis of infrared precipitation estimation is that when the cloud top temperature in a cloud area is below a certain threshold and the extent of the region continues to expand, or the temperature gradient between the cloud top core area and the surrounding cloud area is large, or when the temperature of the cloud area has a downward trend, it indicates the possibility of further development of strong convection and will produce precipitation [23]. There is a limitation in using IR to estimate precipitation, the cloud top motion may not match the precipitation movement [73,74], and this limitation may be one of the reasons why the four satellite-based QPE products do not capture the center of actual heavy precipitation well.
- The details in lines of 551-560:
Interestingly, it can be found in the Oct. 5 NCER (Figure 5) that the IMERG's heavy rainfall centers were concentrated over low-altitude areas (0-1000 m), while GSMaP's heavy rainfall centers were concentrated in high altitude areas (1000-3600 m) and the high-elevation areas received more precipitation than the low-elevation areas. It is possible that the algorithm of GSMaP misclassifies the cold surface on the top of the mountain as rain clouds, resulting in an overestimation of rainfall. As shown in Figure 6, satellite-based QPE products show fair performance (the CC is low, and the RMSE is large). This may be due to the limited spatial resolution of satellite precipitation products (0.1°×0.1°), since precipitation within an area may occur at a scale lower than the satellite pixel size[75].
Point 5: Conclusion: Even if the amount of rain gauge increases over those region, how this will improve the performance of the satellite products?
Response 5: Thanks. We have added explanations and clarifications in the Conclusion section. See lines 660-672 for details. The details are as follows:
In conclusion, we propose that in the future, more ground rainfall stations be used to record actual precipitation data in areas with high spatial heterogeneity and complex terrain in order to evaluate the performance of satellite-based QPE products and avoid the uncertainty in the potential assessment caused by the uncertainty of the reference data. Furthermore, satellite-based QPE products employ IR to measure precipitation, and there may be some mismatch between the cloud top motion and the actual precipitation movement, which may contribute to precipitation misjudgment. We propose that data programmers enhance the retrieval method of satellite-based QPE products for computing weights and make full use of IR-based information. In order to increase the accuracy of satellite-based QPE products, product developers must thoroughly account for the effects of precipitation intensity, sensors, location, and other elements during product development. And the weather forecasters must modify the precipitation prediction in the actual forecast application in view of these variables.
Point 6: L20 “cli-mate”
Response 6: Thanks. We have thoroughly checked the article and corrected some of the errors.
Point 7: L23-24 This sentence does not make sense in the context of the Abstract “The grids contain at least one gauge for both gauge observations and the satellite-based quantitative precipitation estimation (QPE) products.
Response 7: Thanks. We have removed this sentence from the abstract and have modified and added this sentence to the data processing, which can be seen in lines 240-241. The details are as follows:
The gauge observations are compared with the IMERG and GSMaP products for grids that were overlapped with at least one gauge.
Point 8: L39 Any reference for this affirmation? which regions?
Response 8: Thanks. We have added two references based on your suggestions. The references are as follows, and can be seen in lines 45-46.
- The details in lines 45-46 are as follows:
Under a changing climate, both the frequency and intensity of precipitation events tend to increase in many regions[1,2].
- The new references we added are as follow:
Reference:
- Du, H.; Alexander, L.V.; Donat, M.G.; Lippmann, T.; Srivastava, A.; Salinger, J.; Kruger, A.; Choi, G.; He, H.S.; Fujibe, F.; et al. Precipitation From Persistent Extremes Is Increasing in Most Regions and Globally. Ge-ophys. Res. Lett. 2019, 46, 6041–6049, doi:10.1029/2019GL081898.
- Pfahl, S.; O’Gorman, P.A.; Fischer, E.M. Understanding the Regional Pattern of Projected Future Changes in Extreme Precipitation. Nature Clim Change 2017, 7, 423–427, doi:10.1038/nclimate3287.
Point 9: L138 What do you mean by “deeply”?
Response 9: Thanks. It seems to us that “deeply” means analyzing in detail, and perhaps we have a poor wording. We have revised this paragraph, as shown in lines 147-149. The details are as follows:
This study provides a detailed analysis of the uncertainty characteristics of satellite-based QPE products in the context of the Northern China extreme rainstorm in 2021, and evaluates the performance of IMERG and GSMaP satellite-based QPE products.
Point 10: L223. Could you provide some references where this geostatistical method (kriging) was used for precipitation?
Response 10: Thanks. We have revised our paper accordingly based on your comments. See lines 241-243 for details.
- The details in lines 241-243 are as follows:
Kriging interpolation is a technique that is widely used in geoscientific applications such as precipitation [63,64], soils, and ground wind fields.
- The new references we added are as follow:
Reference:
- Prasad, M.S.G. Spatial Prediction of Rainfall Using Universal Kriging Method: A Case Study of Mysuru District. International Journal of Engineering Research 2016, 4, 4.
- Seo, Y.; Kim, S.; Singh, V.P. Estimating Spatial Precipitation Using Regression Kriging and Artificial Neural Network Residual Kriging (RKNNRK) Hybrid Approach. Water Resour Manage 2015, 29, 2189–2204, doi:10.1007/s11269-015-0935-9.

Reviewer 3 Report (New Reviewer)

Author Response
Response to Reviewer 3 Comments
First of all, we are grateful to your helpful suggestions. We accepted all of comments and substantially revised the text. Below, we repeat your comments first and then give our point-by-point response. All responses are in red font for clarity of reading.
Point 1: Why there are many sentences written in red?
Response 1: I'm sorry for giving you a bad reading experience. Since this is a re-submission, I have marked the changes according to reviewers’ comments.
Point 2: Please give references to support the severe precipitation events mentioned in the first paragraph of the Introduction.
Response 2: Thanks. We have added two references based on your suggestions. The references are as follow, and can be seen in lines 45-51. The details are as follows:
Under a changing climate, both the frequency and intensity of precipitation events tend to increase in many regions [1,2]. South Korea and Japan were hit by continuous heavy precipitation in early July 2021, which led to landslides and mass evacuations [3,4]. In mid-July 2021, many countries in western Europe were hit by heavy rains, which caused severe flooding, with the worst affected being Germany, where at least 177 people lost their lives in the floods [5]. Meanwhile, many parts of India and Nepal were affected by heavy precipitation, resulting in many floods and landslides [6].
Reference:
- Du, H.; Alexander, L.V.; Donat, M.G.; Lippmann, T.; Srivastava, A.; Salinger, J.; Kruger, A.; Choi, G.; He, H.S.; Fujibe, F.; et al. Precipitation From Persistent Extremes Is Increasing in Most Regions and Globally. Geophys. Res. Lett. 2019, 46, 6041–6049, doi:10.1029/2019GL081898.
- Pfahl, S.; O’Gorman, P.A.; Fischer, E.M. Understanding the Regional Pattern of Projected Future Changes in Extreme Precipitation. Nature Clim Change 2017, 7, 423–427, doi:10.1038/nclimate3287.
- XINHUANET. Available online: http://www.xinhuanet.com/2021-07/10/c_1127642178.htm (accessed on 5 April 2022).
- Wechat official account. Available online: https://mp.weixin.qq.com/s/RSPj-2Aq5IDvJjunJRROng (accessed on 5 April 2022).
- Beijing News. Available online: https://www.bjnews.com.cn/detail/162747015314135.html (accessed on 5 April 2022).
- Economic and Commercial Office of the Embassy of the People’s Republic of China in Nepal. Available online: http://np.mofcom.gov.cn/article/jmxw/202107/20210703179528.shtml (accessed on 8 April 2022).
Point 3: In the second paragraph of the introduction, the authors discuss the reason for severe precipitation occurring in Henan province. The authors highlight the effect of water vapor on this precipitation. However, authors cannot just simply comment on the reason for this event without any evidence. That’s not the right attitude toward scientific research. Either sufficient scientific proofs (i.e., from observation or simulation) or scientific references should be given to support the idea of the authors.
Response 3: Thanks. Considering the rigor and scientificity of the article, we collected and referred to the literature that can support our position according to your suggestions. The new references we added are as follow and can see in line 58. The details are as follows:
Northern China, especially in Henan Province, mainly affected by typhoon In-Fa and typhoon Cempaka [8,9], has an extreme precipitation event on July 20, 2021.
Reference:
- Xu, H.; Duan, Y.; Li, Y.; Wang, H. Indirect Effects of Binary Typhoons on an Extreme Rainfall Event in Henan Province, China From 19 to 21 July 2021: 2. Numerical Study. JGR Atmospheres 2022, 127, doi:10.1029/2021JD036083.
- Nie, Y.; Sun, J. Moisture Sources and Transport for Extreme Precipitation Over Henan in July 2021. Geophysical Research Letters 2022, 49, doi:10.1029/2021GL097446.
Point 4: L 78-81: The words “in-situ observation” & “indirect estimation” are not professional in the field of precipitation study. The “in-situ observation” & “remote sensing retrieval” should be better. Meanwhile, I don’t think “raindrop distrometer” is an instrument to measure rainfall directly. As I know, the disdrometer (i.e., PARSIVEL, 2DVD) uses the laser to measure the diameter and fall speed of the raindrop, and it has limitations in measuring raindrops at both small and large ends of raindrop diameter. Moreover, it has several assumptions to assume the shape and speed of raindrops, which should be used to retrieve the rain rate. Therefore, its retrieval results have uncertainty to some extent. Please modify this paragraph.
Response 4: Thanks. We have modified them in lines 79-85 based on your suggestions. The details are as follows:
Currently, there are two methods for detecting precipitation: in-situ observation and remote sensing retrieval. In-situ observation instruments include ground rain gauges, and remote sensing retrieval includes ground weather radar and satellite precipitation estimation. In general, the rainfall data obtained by in-situ observation can provide the most accurate precipitation data, but this method does not perform well in terms of spatial continuity. The ground weather radar has a high spatial and temporal resolution, but it’s susceptible to the impact of mountains on detection, resulting in data divergence.
Point 5: Section 2.2.2: The comparison between IMERG and GSMaP algorithm as well as the adopted observations is not easy to understand. Please give a table to summarize the major similarities and differences among the four datasets used in the current manuscript.
Response 5: Thanks for your helpful comments. We give a table to summarize the major similarities and differences among the four datasets used in the current manuscript. See lines 228-233 for details. The details are as follows:
The primary differences and similarities characteristic of satellite-based QPE products that are utilized in this article are summarized in Table 1.
|
Product |
Spatial / Temporal Resolution |
Spatial Domain |
Main Input Data |
Latency |
Applications |
|
IMERG_ER |
0.1°/ 0.5 hour |
90°N-90°S |
PMW, DPR1, GMI2, PR3, TMI4, AMSR25, SSMIS6 |
4 hours |
Prediction of flash floods and precipitation |
|
IMERG_LR |
0.1°/ 0.5 hour |
90°N-90°S |
PMW, DPR, GMI, PR, TMI, AMSR2, SSMIS |
14 hours |
Water resource management |
|
GSMaP_NRT |
0.1°/ 1 hour |
60°N-60°S |
TMI, GMI, SSMIS, PR, DPR, AMSU-A7/ AMSU-B8 |
4 hours |
Real-time applications |
|
GSMaP_MVK |
0.1°/ 1 hour |
60°N-60°S |
TMI, GMI, SSMIS, PR, DPR, AMSU-A/ AMSU-B |
3 days |
Water resource management |
Table 1. The major differences and common features of satellite-based QPE products.
1 GPM Dual-frequency Precipitation Radar; 2 GPM Microwave Imager; 3 TRMM Precipitation Radar; 4 TRMM Microwave Imager; 5 Advanced Microwave Scanning Radiometer Version 2; 6 Special Sensor Microwave Imager/Sounder; 7 Advanced Microwave Sounding Unit A; 8 Advanced Microwave Sounding Unit B.
Point 6: L291-292: According to Figure 3, the rainfall amount in the eastern coastal area of Shandong Province is highly overestimated by GSMaP than that by IMERG, which means the GSMaP is worse than the. IMERG for this case. So, why do authors say the GSMaP “outperform” the IMERG? Please explain it.
Response 6: Thanks for comments. The phrase has been altered on lines 311-313 as follows:
GSMaP products continue to overestimate precipitation in the eastern coastline area of Shandong Province when compared to IMERG data in this sub-precipitation center.
Point 7: L365-367: If this is a unique finding by the authors, please give proof to support your point (the importance of water vapor transmission in this case).
Response 7: Thank you for your helpful comments. After analyzing the relevant literature, we discovered evidence that supported our initial assumptions. The new references we added are as follow and can see in lines 384-386. The details are as follows:
This might be mainly because typhoon In-Fa and typhoon Cempaka did not directly affect the weather in the Northern China but instead transferred water vapor to Northern China through atmospheric circulation[9].
Reference:
- Nie, Y.; Sun, J. Moisture Sources and Transport for Extreme Precipitation Over Henan in July 2021. Geophysical Research Letters 2022, 49, doi:10.1029/2021GL097446.
Point 8: L497-501: Authors quote a news report to support their point of synoptic conditions. The news report was non-professional and without peer-reviewed. It cannot be used as scientific proof in a professional journal. Please replace it with a professional reference.
Response 8: Thank you for your helpful comments. Considering the rigor and thoroughness of the study, we could not obtain the appropriate references to explain the meteorological conditions of the first severe precipitation event and the third extreme precipitation event, therefore we chose to remove them.
Point 9: As the authors describe: there are 906 rainfall stations in Northern China as shown in Figure 1b. I doubt why the Jiangsu & Anhui Provinces can be involved in Northern China. However, there are rain gauge stations in these two provinces noted on the map. Please give the definition of Northern China.
Response 9: Thanks to the reviewer for your ideas. We examine this since we picked three severe precipitation cases, although the affected range of each precipitation event is different. We want to thoroughly depict the breadth of effect of the three precipitation events, and increase the latitude and longitude, so portions of Jiangsu and Anhui are included. At the same time, because the precipitation data in these two locations has no influence on the real precipitation data, the data is preserved, and the retained rainfall stations solely serve as a display. We define northern China as the region impacted by three extreme precipitation events, see study area for details in lines 163-167. The details are as follows:
The study area locates in northern China, which are affected by 12 Jul. 2021 Northern China Extreme Rainstorm, 20 Jul. 2021 Northern China Extreme Rainstorm, and 5 Oct. 2022 Northern China Extreme Rainstorm in 2021 (hereafter referred to as Jul. 12 NCER, Jul. 20 NCER and Oct. 5 NCER), spanning from 31°N to 43°N in latitude and from 105°E to 123°E in longitude (Figure 1).
Point 10: Section 3.2: How many rain gauge stations participated in the calculation of Mean Hourly rainfall for each case? There are different regions shown in Figures 3-5. There are also significantly different precipitation centers among the three cases. However, the authors don’t give the number of stations for the calculation of Mean Hourly rainfall, which has a great impact on the results. If the authors use the total 906 rainfall stations shown in Figure 1b for the calculation, is it possible that the rain gauges with low rain rates or non-rainfall could lower the Mean Hourly rainfall or even distort the trend of Mean Hourly rainfall? On the other side, if the authors use part of the 906 stations for the calculation, how to select the rain gauges objectively? If cannot do the rain gauge selection objectively, how to apply the satellite-based QPE for the operational work, such as the authors mentioned “prediction”? Please explain.
Response 10: Thanks. Sorry for our expression problems, resulting in ambiguity in the statement. In reality, the rainfall station data utilized in each precipitation event is different. We have supplemented the rainfall stations utilized for each precipitation event in the study area. See lines 183-185 for details. The details are as follows:
In this paper, the number of surface observation stations employed in three extreme precipitation occurrences varies: 557 on Jul. 12 NCER, 535 on Jul. 20 NCER, and 831 on Oct. 5 NCER.
Point 11: Figure 8c: In this case, the variation trend of the rain rate estimated by satellites is almost opposite to the observations from rain gauges within54-90 hours. Please explain the reason.
Response 11: Thanks. We have added explanatory notes in the Discussion. See lines 575-581 for details. The details are as follows:
In view of this scenario of satellite-based QPE products, we plotted the hourly spatial distribution maps of three extreme precipitation events. According to the spatial distribution maps, satellites typically fail to detect or underestimate precipitation in places with high elevations and complicated terrain, and even overestimate precipitation in low-altitude (0-1000 m) and flat areas. This demonstrates that altitude and terrain complexity continue to be major variables influencing the accuracy of satellite-based QPE products.
Point 12: The discussions and conclusions of temporal characteristics of four satellite-based products are mainly based on the analysis of section 3.2. Therefore, this part of the conclusions may be not believable if the authors cannot explain the comments of 10 and 11 reasonably.
Response 12: Thank you for your helpful comments. We have modified and added explanations. See Response 10 and Response 11 for details.
Point 13: How to define “Low-altitude”? There are many places that mentioned this word without the definition. Please explain it.
Response 13: Thanks. We have marked certain portions of the article, but not entirely marked. We have now properly marked. Here we say the low altitude is 0-1000 m. See lines 30, 323, 517, and 552 for details. The details are as follows:
- The details in lines of 30:
The IMERG products capture strong precipitation centers that are compatible with the gauge observations, especially in extreme precipitation events in areas with relatively flat terrain and low-altitude (0–1000 m).
- The details in lines of 323:
The Jul. 20 NCER heavy rainfall event has both long duration and high intensity, and it produces a variety of meteorological calamities in low-altitude (0-1000 m) areas of central and northwestern Henan Province.
- The details in lines of 517:
As shown in Figure 3 and Figure 5, it can be found that both IMERG products and GSMaP products overestimated precipitation in low-altitude (0-1000 m) areas during the Jul. 12 NCER and Oct. 5 NCER, which are extreme precipitation occurrences induced by non-typhoons.
- The details in lines of 552:
Interestingly, it can be found in the Oct. 5 NCER (Figure 5) that the IMERG's heavy rainfall centers were concentrated over low-altitude areas (0-1000 m), while GSMaP ' s heavy rainfall centers were concentrated in high-altitude areas (1000-3600 m) and the high-elevation areas received more precipitation than the low-elevation areas.
Point 14: As mentioned by the authors in Section 2.2.2, there are at least 4 hours delays for the IMERG and GSMaP products. Meanwhile, there are only 2-4 hours in advance for these products to estimate the peak of precipitation (L573). So how to use the satellite-based precipitation products for the prediction of the peak of extreme precipitation? Don’t the authors think the occurrence of the peak of precipitation would be prior to the estimation of the satellite-based products? I even haven’t considered the time consumption of data download and analysis. Please explain.
Response 14: Thanks. Satellite-based QPE products are real-time and could be accessed in real-time just like radar observation data. But the study data that supplied the analysis was postponed until today. According to the assessment of the products, we may utilize the necessary research findings to pick the satellite products in the prediction.
Point 15: There are only three cases in the current study. Therefore, the conclusion may be not believable, especially for “the GSMaP products perform more effectively in non-typhoon events, while IMERG products behave well in typhoon events”. Since the GPM project begins in 2014, it is possible for the authors to do research on more extreme rainfall cases influenced by typhoons and non-typhoons to support their ideas.
Response 15: Thank you for your helpful comments. Sorry, this one is we careless, forgot to include qualifiers, we have amended the appropriate information. See line 637 for details. The details are as follows:
3) In terms of POD, CSI, and FAR performance in extreme precipitation events, the GSMaP products perform more effectively in non-typhoon extreme precipitation events (Jul. 12 NCER and Oct. 5 NCER), whereas IMERG products behave well in typhoon extreme precipitation events (Jul. 20 NCER).
Point 16: L20: Please check words like “cli-mate” throughout the manuscript.
Response 16: Thanks. We have modified and checked the full paper.
Point 17: L85-87: To my knowledge, the spatiotemporal resolution of precipitation products from satellites is lower than those from ground weather radar. Is it suitable to use “higher” spatial and temporal resolution here?
Response 17: Thank you for your helpful comments. I 'm sorry, there is ambiguity here because of the wrong expression. We mean that a satellite has higher spatial and temporal resolution than in-situ observation, and a satellite has better continuity than radar. We modified the statements. See lines 85-90 for details. The details are as follows:
Satellite-based QPE products provide greater coverage than radar quantitative precipitation measurement, making them particularly useful for precipitation monitoring in mountainous locations, deserts, and seas. Due to the advantages of a large detection range, strong spatial continuity, and high spatial and temporal resolution, the satellite can quickly capture mesoscale heavy rainfall.
Point 18: L135: These matrices are mentioned for the first time in the main content. Please give them their full names.
Response 18: Thanks. We have modified and added the full names. See lines 139-144 for details. The details are as follows:
For these extreme precipitation events in Northern China, the study takes the ground rain gauge observations as a reference, and adopts the Correlation Coefficient (CC), Relative Bias (RB), Root-Mean-Squared Error (RMSE), and Fractional Standard Error (FSE), as well as the Probability of Detection (POD), False Alarm Ration (FAR) and Critical Success Index (CSI) to evaluate data from the Version 06 of the GPM mission developed by NASA IMERG and a new satellite precipitation product developed by JAXA the GSMaP in Version 07.
Point 19: L246: What does the “rain gauge readings” stands for?
Response 19: Thanks. We are here to express the meaning of rain gauge data, but seem to produce ambiguity. Now, we have modified it. See lines 265-266 for details. The details are as follows:
The spatial and temporal correlation between satellite precipitation products and rain gauges is represented by CC.

Round 2
Reviewer 1 Report (Previous Reviewer 1)
REVIEW of the revised version of the manuscript "Evaluation and Uncertainty Analysis of IMERG and GSMaP Satellite Precipitation Products in Northern China" by Huiqin Zhu, Sheng Chen, Zhi Li, Liang Gao, Xiaoyu Li [Title. Remote Sens. 2022,14, x. https://doi.org/10.3390/xxxxx].
The new version of the manuscript is significantly improved compared to the former version. The authors have been taken into account most of my comments. Because of that, I think that this version of the manuscript can be published in this eminent journal.
Author Response
Thanks for your helpful comments!Have a good day!
Reviewer 2 Report (Previous Reviewer 2)
Accepted
Author Response
Thanks for your helpful comments!Have a good day!
Reviewer 3 Report (New Reviewer)
This revised manuscript can be published.
Meanwhile, the number of rain gauges used for each case in the current manuscript is subjective and should be improved in the future. More cases should be studied also.
Author Response
Response to Reviewer 3 Comments
Point 1: Meanwhile, the number of rain gauges used for each case in the current manuscript is subjective and should be improved in the future. More cases should be studied also.
Response 1: Thanks for your helpful comments!We agree with your viewpoint. We also highlighted in the Discussion (lines 609-617) and Conclusion (lines 662-664) of the paper that the number and spatial distribution of rainfall stations would impact the accuracy of the evaluation. We only utilized accessible national station rainfall data in this research since we couldn't get rainfall station data from other provincial regional stations, which makes us unable to make a more detailed and accurate assessment. If we can receive rainfall data from the regional station, we will do a further in-depth analysis. Simultaneously, we will investigate including more years of extreme precipitation events in future research. Thanks for your comments on our article!
- The details in lines 609-617 are as follows:
Although we want to evaluate the performance of satellite-based QPE products more accurately due to the current inability to obtain regional rain gauge observations in various provinces, we can only use national station rainfall data for research. However, scholars interested in satellite-based QPE products evaluation can consider using the regional rain gauge observations for research. At the same time, we feel that the current national station rainfall stations are mostly located in flat terrain and low-altitude (0-1000 m) locations, making it impossible to assess the effectiveness of satellite-based QPE products in high-altitude mountainous areas, particularly for snowfall event monitoring.
- The details in lines 662-664 are as follows:
Nowadays, some studies [13] have found that the distribution density of ground rainfall stations is also one of the important factors affecting the satellite precipitation assessment.
Reference:
- Wang, H.; Shao, Z.; Gao, T.; Zou, T.; Liu, J.; Yuan, H. Extreme Precipitation Event over the Yellow Sea Western Coast: Is There a Trend? Quaternary International 2017, 441, 1–17, doi:10.1016/j.quaint.2016.08.014.

This manuscript is a resubmission of an earlier submission. The following is a list of the peer review reports and author responses from that submission.
Round 1
Reviewer 1 Report
REVIEW of the manuscript "Assessing the GPM Satellite Precipitation Products during Oc-2 tober 2021 Shanxi Extreme Rainstorms" by Huiqin Zhu, Sheng Chen, Zhi Li,Liang Gao [Remote Sens. 2022,14, x. https://doi.org/10.3390/xxxxx ].
|
|
This paper presents the performance of Integrated Multi-satellitE Retrievals for Global Precipitation Measurement (GPM) mission (IMERG) and Global Satellite Mapping of Precipitation (GSMaP) quantitative precipitation estimation (QPE) products over Shanxi province of China during October 2021 using ground-based rainfall station observation data for comparisons. The authors used the various indicators for the agreement of the proposed method data with referent precipitation data. The presented results of comparisons are promising from the point of view of its further use.
The topics of this manuscript is very important and may be of the great scientific and practical interest. Because of that I am confident that this paper will be acceptable for publication after a major revision.
Comments
The abstract is too based on enumerating the results. The abstract should also contain the novelty of the method and its advantage over others used.
The Introduction section should have one paragraph with a review of the literature outside China about the topics of this manuscript. There should also be a critical review of the shortcomings of the methods used.
Subsection 2,3: What statistical metrics were used by the other authors (not only WMO)?
The analysis of the results is too based on enumeration of results. Such an analysis must provide deep insight why the proposed method almost never agrees with the reference values and suggest what to do next on this issue. This is important from the point of view of the further application of the proposed method.
The conclusion must explicitly emphasize the novelty of the method, its advantage over others used and make suggestions for its improvement. Only then, it would be of great practical and scientific interest.
Lines 202-204: Explain this claim in more detail.
Fig.1 Distances along the meridian and in the zonal direction are not the same. Let's correct this.
Figs 2 and 3 need to have also distances (in km) in both directions because they are not equal.
Reviewer 2 Report
This manuscript shows a number of clear results, and contains ample discussion of them. I recommend the manuscript be accepted.
Reviewer 3 Report
This manuscript assesses the GPM products during Oct 2021 Shanxi rain storm. This is a very small case study and lacks of enough new findings about the satellite precipitation retrievals. It is well known that plenty of papers have been published about the evaluation of satellite precipitation estimates (no merely GPM) in China, and I don't think it is ready for publication in Remote Sensing. I would like to reject this paper. Some of my primary concerns are below:
1. Please perform a comprehensive evaluation of all types of satellite precipitation estimates (e.g., GPM IMERG, CMROPH, TMPA, GSMaP, PERSIANN-era) during several typical severe weather extremes (e.g., hail, hurricanes, etc) in China or a larger region. The evaluation approaches are not limited in statistical analysis and also extended to hydrologic evaluation.
2. In the discussion, please provide a solution about the improvement of the current satellite precipitation estimates in the weather extremes.